# A novel inhibitory nucleo-cortical circuit controls cerebellar Golgi cell activity

Lea Ankri[1†], Zoé Husson[2,3,4†], Katarzyna Pietrajtis[2,3,4], Rémi Proville[3,4,5], Clément Léna[3,4,5], Yosef Yarom[1], Stéphane Dieudonné[2,3,4*], Marylka Yoe Uusisaari[1*]

[1]Department of Neurobiology, Edmond and Lily Safra Center for Brain Sciences, Hebrew University of Jerusalem, Jerusalem, Israel; [2]Inhibitory Transmission Team, Institut de Biologie de l'École Normale Supérieure, Ecole Normale Supérieure, Paris, France; [3]Centre national de la recherche scientifique, Institut de Biologie de l'École Normale Supérieure, Ecole Normale Supérieure, Paris, France; [4]Institut national de la santé et de la recherche médicale, Institut de Biologie de l'École Normale Supérieure, Ecole Normale Supérieure, Paris, France; [5]Cerebellum Team, Institut de Biologie de l'École Normale Supérieure, Ecole Normale Supérieure, Paris, France

**Abstract** The cerebellum, a crucial center for motor coordination, is composed of a cortex and several nuclei. The main mode of interaction between these two parts is considered to be formed by the inhibitory control of the nuclei by cortical Purkinje neurons. We now amend this view by showing that inhibitory GABA-glycinergic neurons of the cerebellar nuclei (CN) project profusely into the cerebellar cortex, where they make synaptic contacts on a GABAergic subpopulation of cerebellar Golgi cells. These spontaneously firing Golgi cells are inhibited by optogenetic activation of the inhibitory nucleo-cortical fibers both in vitro and in vivo. Our data suggest that the CN may contribute to the functional recruitment of the cerebellar cortex by decreasing Golgi cell inhibition onto granule cells.

*For correspondence: dieudon@biologie.ens.fr (SD); yoe@iki.fi (MYU)

†These authors contributed equally to this work

Competing interests: The authors declare that no competing interests exist.

## Introduction

The cerebellum plays a key role in the fine temporal control of posture and movements as well as in cognitive processes (*Ito, 1993*; *Leiner et al., 1993*). Current cerebellar theories (*Apps and Garwicz, 2005*; *Jacobson et al., 2008*; *Dean and Porrill, 2011*) mainly discuss cerebellar computation from the point of view of its cortical circuitry, where both pre-cerebellar mossy fibers (MFs) and inferior olive (IO)-originating climbing fibers (CFs) modulate Purkinje neuron (PN) spiking. Sensory-motor signal processing in the main cerebellar output structure, the cerebellar nuclei (CN), has received less attention. Modulation of spike frequency and timing in the CN projection neurons is considered to be mostly determined by the massive inhibitory cortico-nuclear projection of PNs (*Chan-Palay, 1977*; *De Zeeuw and Berrebi, 1995*; *Gauck and Jaeger, 2000*; *Telgkamp and Raman, 2002*; *Pedroarena and Schwarz, 2003*; *Telgkamp et al., 2004*; *Person and Raman, 2012*; *Najac and Raman, 2015*). However, certain aspects of cerebellar function persist even when the cerebellar cortex is selectively inactivated or damaged (*Thompson and Steinmetz, 2009*; *Aoki et al., 2014*; *Clopath et al., 2014*; *Longley and Yeo, 2014*). Thus, a better understanding of the information processing in the CN as well as its influence on cerebellar cortical computation is needed.

Currently, the cerebellar cortex and the CN are known to interact through two circuits. The best known is the nucleo-olivary (NO) circuit (*Apps and Garwicz, 2005*; *Apps and Hawkes, 2009*; *Chaumont et al., 2013*), where the small GABAergic CN cells, subject to PN inhibition (*Najac and Raman, 2015*), project to the contralateral IO (*Fredette and Mugnaini, 1991*). This pathway regulates

**eLife digest** The cerebellum is a region in the brain that plays a central role in controlling posture and movement. The cerebellum is composed of a cortex and several nuclei. The nuclei are thought to 'compute' the signals that are sent from the cerebellum to other parts of the brain to control posture and movement. They do this under the supervision of the cortex. The main interaction between the cortex and the nuclei involves cortical neurons called Purkinje cells inhibiting the activity of the nuclei.

Ankri, Husson et al. have now used various genetic techniques and mutant mice to identify a new population of neurons in the nuclei of the cerebellum and to express fluorescent markers into these cells. This approach reveals that the axons of these neurons 'climb' from the nuclei to the cortex to form a new circuit called the inhibitory nucleo-cortical (iNC) pathway. Moreover, activating the iNC axons with light reveals that they selectively target and silence a population of neurons called the Golgi cells, which control the transmission of information in the cerebellar cortex.

Ankri, Husson et al. go on to show that the Golgi cells silenced by the iNC pathway differ from other Golgi cells in a number of ways: in particular, these Golgi cells use a chemical called GABA to communicate with neurons. The next challenge is to explore how the iNC pathway fine-tunes how sensory inputs are processed in the cerebellum, and to better understand its role in the execution of complex movements, including in people with conditions that affect motor function, such as ataxias.

olivary activity (*Chen et al., 2010*; *Bazzigaluppi et al., 2012*; *Chaumont et al., 2013*; *Lefler et al., 2014*) and thereby complex spike activity in the PNs and cerebellar cortical plasticity (*Hansel and Linden, 2000*; *Coesmans et al., 2004*; *Bengtsson and Hesslow, 2006*; *Medina and Lisberger, 2008*). A less-known nucleo-cortical circuit is formed by the glutamatergic neurons of the CN which, in addition to projecting to various premotor and associative regions of the brain (*Tsukahara and Bando, 1970*; *Asanuma et al., 1980*; *Angaut et al., 1985*; *Sultan et al., 2012*; *Ruigrok and Teune, 2014*), send axonal collaterals to the cerebellar granule cell layer (GrCL; *Houck and Person, 2015*). These collateral fibers form MF-like terminals contacting granule cell (GrC) and Golgi cell dendrites (see also *Tolbert et al., 1976*, *1977*, *1978*; *Hámori et al., 1980*; *Payne, 1983*). The functional significance of this excitatory nucleo-cortical (eNC) pathway, loosely following the modular arrangement of the cerebellum (*Dietrichs and Walberg, 1979*; *Gould, 1979*; *Haines and Pearson, 1979*; *Tolbert and Bantli, 1979*; *Buisseret-Delmas, 1988*; *Provini et al., 1998*; *Ruigrok, 2010* ; reviewed by *Houck and Person, 2013*), is likely related to efference copying of motor commands to the cerebellar cortex (*Sommer and Wurtz, 2008*; *Houck and Person, 2015*).

In addition to the pathways linking the CN with the cerebellar cortex mentioned above, evidence has occasionally emerged for an inhibitory nucleo-cortical (iNC) pathway. GABAergic neurons have been shown to be labeled in the CN by retrograde tracing from the cerebellar cortex (*Batini et al., 1989*), and nucleo-cortical terminals with non-glutamatergic ultrastructural features have been found to contact putative Golgi cell dendrites (*Tolbert et al., 1980*). More recently, it was demonstrated that GlyT2-expressing CN neurons extend axons toward the cerebellar cortex (*Uusisaari and Knöpfel, 2010*), suggesting that the iNC pathway might be identifiable by its glycinergic phenotype. While the iNC projection is likely to have significant impact on cerebellar computation, its postsynaptic targets and its functional organization remain unknown.

To establish the existence and prevalence of an inhibitory connection between the CN and the cerebellar cortex, we employed specific viral targeting of GABAergic and glycinergic neurons in the CN of GAD-cre and GlyT2-cre transgenic mouse lines, respectively (*Taniguchi et al., 2011*; *Husson et al., 2014*). We found that the GABA-glycinergic CN neurons form an extensive plexus of iNC axons, which contact Golgi cells in the cerebellar granular and molecular layers. Specific optogenetic activation of the iNC axons inhibited spikes in a distinct subpopulation of Golgi cells, characterized by their spontaneous firing, high neurogranin immunoreactivity, and negligible GlyT2 expression. As the functional significance of the iNC pathway is likely to be amplified by the high divergence of Golgi cells, which target thousands of GrCs (*Hámori and Somogyi, 1983*; *Jakab and Hamori, 1988*; *Andersen et al., 1992*; *Korbo et al., 1993*), as well as the remarkable mediolateral extent of the iNC axons, the CN might play a key role in the regulation of the information flow through the GrCL.

## Results

### Nucleo-cortical projection neurons have a mixed GABA-glycine phenotype

In order to identify the iNC projection neurons, we specifically labeled the GABAergic and glycinergic CN neurons by injecting floxed adeno-associated virus (AAV) in the CN of GAD-cre and GlyT2-cre transgenic mouse lines, respectively. As shown in *Figure 1A1,B1*, these procedures resulted in the expression of the fluorophores (mCherry in GAD-cre and YFP in GlyT2-cre mice) in a subset of CN neurons. In the GAD-cre mice, the labeled neurons displayed a wide range of sizes and shapes, including both globular and multipolar morphologies (*Figure 1A2*, arrow and arrowhead, respectively). In contrast, in GlyT2-cre mice, the labeled neurons were predominantly large (*Figure 1B2*, arrowhead) and multipolar, often with a thick principal dendrite (*Figure 1B2*, arrows). To examine the morphological difference between CN cells labeled in GAD-cre and GlyT2-cre mice, we measured and compared their soma sizes. The size distribution in GAD-cre CN was best fitted with a two-component Gaussian model (*Figure 1D*, red bars and line; Gaussian peaks at 11.9 μm and 16.2 μm; R-square 0.97, n = 650 cells in 6 animals), suggesting it is composed of two separate populations. In contrast, the optimal fit to the size distribution of GlyT2-cre neurons was obtained with a single-component Gaussian model (*Figure 1D*, yellow bars and line; peak at 16.6 μm, R-square = 0.83, n = 118 cells in 4 animals; KS-test GAD vs GlyT2, $p < 0.0001$). The peak of this GlyT2-fit matched well with the right-most peak in the GAD-cre distribution (GAD-cre, second peak confidence interval, 13.6–18.8 μm; GlyT2-cre confidence interval, 16.0–17.1 μm).

The difference between the GAD-cre and GlyT2-cre populations, corresponding to the left-most peak in the GAD-cre distribution (*Figure 1D*), likely corresponds to the NO cells that are also transfected in the GAD-cre model, as evidenced by the presence of fluorescent axons in the IO (*Figure 1A3*; see *Lefler et al., 2014*). To confirm this, we retrogradely labeled the NO cells via viral injections in the IO (*Figure 1C*). The size distribution of the NO neurons (mean: 12.8 ± 2.4 μm; n = 193 cells in 4 animals; see also *Najac and Raman, 2015*) was significantly different from the GlyT2 cells (NO vs GlyT2 KS-test, $p < 0.0001$, *Figure 1D*). Furthermore, the NO size distribution was well fitted with a single Gaussian with a peak closely resembling the left-most peak of the GAD-cre distribution (*Figure 1D*, green bars and line; peak at 12.3 μm, confidence interval 12.0–12.7 μm; R-square 0.93; n = 193 cells in 4 animals). Thus, we conclude that the mixed GABA-glycinergic neurons form a separate population from the purely GABAergic NO neurons that are not transfected in adult GlyT2-cre animals (*Husson et al., 2014*). These glycinergic neurons, like all other CN neurons, receive functional inputs from PN axons (*Figure 1—figure supplement 1*), as previously suggested by immunohistochemical and optogenetic studies (*De Zeeuw and Berrebi, 1995*; *Teune et al., 1998*).

In contrast to the NO axons, which leave the CN towards the brainstem, we found that axons of the large multipolar GAD and GlyT2-positive neurons projected across the white matter surrounding the CN and into the cerebellar cortex (as shown in *Figure 1E1,2* for the GAD-cre and GlyT2-cre cerebella, respectively). In the vermis, the projections regularly crossed the midline and extended into the contralateral cortex, but otherwise the projection was predominantly ipsilateral. The divergence of nucleo-cortical axons in the cortex varied depending on the extent and localization of viral transfection, coarsely following the known cerebellar modules (*Apps and Garwicz, 2005*; *Pijpers et al., 2005*; *Apps and Hawkes, 2009*). Lateral CN injections labeled axons in the lateral and intermediate hemispheres and the floccули (*Figure 1F1*), whereas medial CN injections yielded labeled axons predominantly in the vermal cerebellum (*Figure 1F2*). Surprisingly, individual nucleo-cortical axons could be seen to travel long distances in the medio-lateral direction (up to several millimeters; see inset in *Figure 1F2*) forming boutons within the GrCL.

The nucleo-cortical axons formed dense meshes in the GrCL (*Figure 2A1,B1*). As seen in high-magnification images (*Figure 2A2,B2*), the axons formed large swellings that were also seen in the molecular layer (ML; *Figure 2A3,B3*). The axons in the two cre lines were remarkably similar in their appearance, even though the swellings labeled with mCherry in the GAD-cre line appeared nearly identical to the varicosities labeled by YFP in the GlyT2-cre line (*Figure 2A2,B2,C1*; cross-sectional areas in GAD-cre, red bars, 2.1 ± 0.9 μm², n = 400 varicosities; in GlyT2-cre, yellow bars, 1.87 ± 0.9 μm², n = 415 varicosities; KS-test, $p = 0.013$). Also, no large differences were evident among boutons found in the GrCL or ML (*Figure 2C2*; KS-test, $p = 0.023$). These anatomical similarities imply that the axons

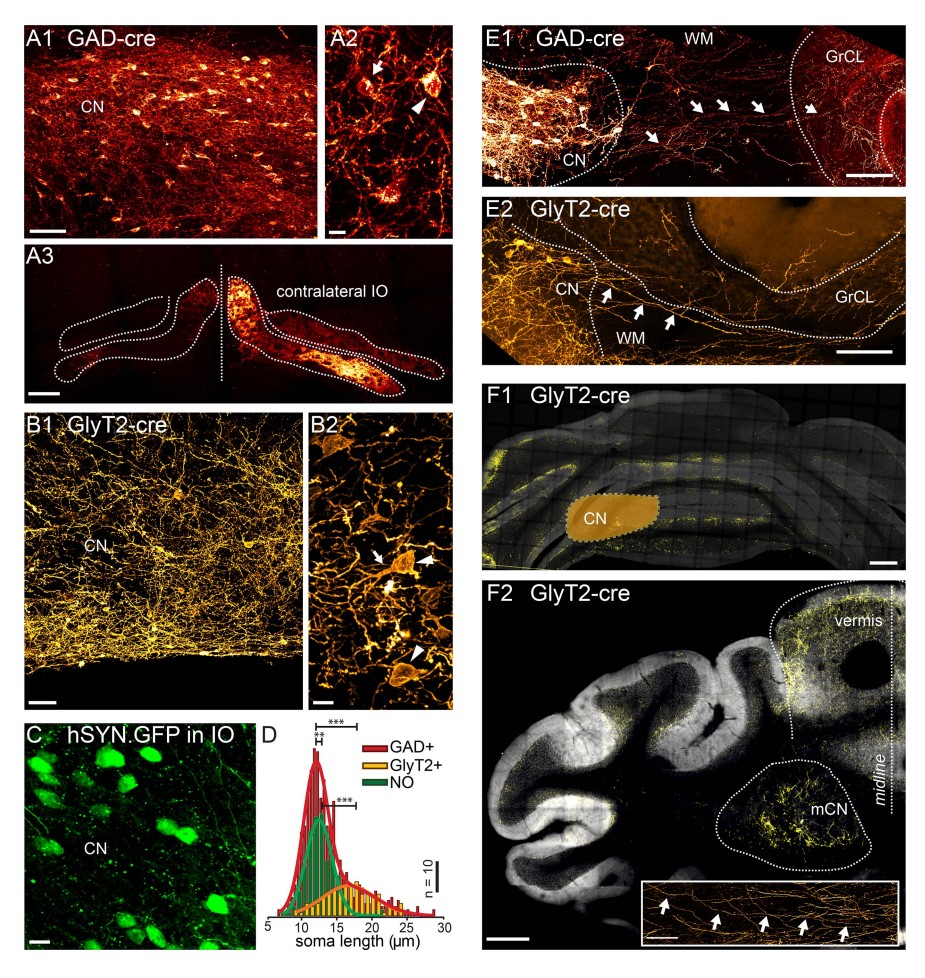

**Figure 1.** Targeted viral labeling of the GABAergic and glycinergic neurons in the CN reveals dense and wide-spread network of nucleo-cortical axons in the cerebellar cortex. (**A1–B1**) Confocal images of coronal cerebellar sections in mice, where floxed virus was injected into the cerebellar nuclei of GAD-cre and GlyT2-cre mice. (**A1**) Flocculus, coronal view, 40× confocal scan tiles. (**B1**) Posterior vermis, horizontal view. (**A2–B2**) Higher magnification confocal images of the cerebellar nuclei show transfected GABAergic and glycinergic neurons, respectively. Note the lower density of labeled neurons in GlyT2-cre brain. In the GAD-cre mice, both small, globular and larger, multipolar neurons (arrows and arrowheads in **A2**, respectively) were seen. In the GlyT2-cre mice, only large, multipolar neurons were observed, characteristic to the glycinergic CN neurons (arrowheads in **B2**). In the GAD-cre mice, the transfected neurons included the GABAergic NO neurons, as evidenced by the fluorescent axons in the contralateral IO (**A3**). (**C**) Injection of hSyn-GFP-virus into the IO retrogradely labeled small, round NO neurons in the contralateral CN. (**D**) Comparison of soma sizes among the three labeled populations. The histograms are fitted with single (GlyT2+ and NO) or double (GAD+) Gaussians (thick lines), showing that the GlyT2+ neurons distribution matches the second peak in GAD + fit and that the GlyT2+ and NO neurons form distinct populations that contribute to the GAD + population. (**E**) Confocal composite images showing virally transfected CN neurons in GAD-cre (**E1**) and GlyT2-cre (**E2**) mice, and their axons (arrows) extending across the WM surrounding the CN into the GrCL of the cerebellar cortex. (**F1**) Confocal composite image of a caudal coronal section of GlyT2-cre cerebellum where the lateral CN was virally transfected (location of the CN is drawn schematically on top of the image). The wide distribution of the NC axons in medio-lateral direction, including parts of the contralateral cerebellum, is shown in yellow color. (**F2**) Confocal composite image of a horizontal section at the level of the CN in GlyT2-cre cerebellum with transfection of the glycinergic neurons in the medial CN. The axons of the labeled neurons can be seen extending through wide areas of the vermal cortex. The inset shows a single inhibitory nucleo-cortical (iNC) axon forming axonal swellings across several hundreds of μm in the GrCL. Abbreviations: GrCL, granule cell layer; CN, cerebellar nuclei; mCN, medial cerebellar nuclei; IO, inferior olive; NO, nucleo-olivary; WM,

*Figure 1. continued on next page*

*Figure 1. Continued*

white matter. Scale bars: **A1**, **B1**: 50 µm; **A2**, **B2**, **C**, 10 µm; **E**, 100 µm, **F1**, **2**: 400 µm; **F2**, inset: 100 µm. See also *Figure 1—figure supplement 1*.

The following figure supplement is available for figure 1:

**Figure supplement 1**. iNC neurons receive functional GABAergic Purkinje Neuron inputs.

labeled in the two transfection models represent the projections of a specific population of CN neurons with a mixed GABA-glycinergic phenotype (*Husson et al., 2014*). Indeed, immunostaining revealed that virtually all the nucleo-cortical fibers in the GAD-cre transfected mice were immunoreactive for GlyT2 (*Figure 2D1–2*; 94.6 ± 6.2%, n = 2 animals, n = 9 stacks, n = 422 varicosities; *Figure 2D*), while those in GlyT2-cre cerebella were reactive for GAD65-67 (93.9 ± 5.0%, n = 3 animals, n = 7 stacks, n = 565 varicosities; *Figure 2E1–2*). These results unequivocally demonstrate the dual neurotransmitter phenotype of the nucleo-cortical projection. Notably, neither rosette-like terminals nor evidence of contacts within cerebellar glomeruli was found. This indicates that they differ both in shape and location from the excitatory MFs and the glutamatergic nucleo-cortical fibers described earlier in the literature, both forming rosette-like terminals within the glomeruli (*Tolbert et al., 1978*; *Hámori et al., 1980*; *Batini et al., 1992*; *Houck and Person, 2015*).

## Nucleo-cortical fibers inhibit Golgi cell activity

Having demonstrated the existence of a GABA-glycinergic projection from the CN to the cerebellar cortex, generated by a distinct cell type of the CN, we proceeded to identify the targets of this iNC pathway. Golgi cells, which are the only ubiquitous cerebellar neurons that express glycine receptors in the cerebellar cortex (*Dieudonné, 1995*), as well as the only neurons with dendrites both in the granular and molecular layers, constitute the most likely targets for iNC axons. To investigate this possibility, we introduced a non-specific GFP-expressing virus to the cerebellar cortex of GAD-cre mice transfected as above in the CN, to be able to visualize neurons in the GrCL. This procedure labeled Golgi cells and indeed we found axonal swellings of iNC fibers apposed along the proximal dendrites and cell bodies of Golgi cells (*Figure 3A*, arrows).

To confirm the presence of functional inhibitory synaptic connections between the CN and the cerebellar cortex, we selectively activated channelrhodopsin 2 (ChR2) expressed in the iNC axons in acute slices (as shown schematically in *Figure 3B*, left panel). First, we performed voltage-clamp whole-cell recordings from Golgi cells surrounded by transfected iNC fibers in GlyT2-cre mice (*Figure 3B*, right panel). With the use of small collimated beams of light (see 'Materials and methods'), we stimulated locations near Golgi cell dendrites with a single, short (5 ms) pulses. Inhibitory post-synaptic currents (IPSCs) were evoked in 9 out of 38 recorded Golgi cells (23.7%; *Figure 3C1*) with a mean amplitude of 40 ± 28 pA. Given the ionic composition of our experimental solutions, the estimated reversal potential of −74 mV with the permeabilities of bicarbonate and chloride taken into account and the holding potential of −50 mV, the chord synaptic conductance was 1.7 ± 1.2 nS and the slope conductance was 1.9 ± 1.3 nS according to the Goldman–Hodgkin–Katz (GHK) equation. We calculated that the equivalent peak conductance of the iNC synapse measured in symmetrical chloride conditions would have been of 8.5 ± 5.9 nS. The light-evoked IPSCs had a 10–90% rise time of 2.5 ± 1.3 ms and a bi-exponential decay ($\tau1 = 8.2 \pm 1.9$ ms, 52.3 ± 18.3%; $\tau2 = 34.8 \pm 9.2$ ms, n = 9 cells). Application of strychnine at a concentration selective for glycine receptors (300 nM) decreased the amplitude of the IPSC by 24 ± 25% (p = 0.039, n = 9 cells) without affecting the time course of the IPSCs (rise time: 2.4 ± 0.9 ms; decay: $\tau1 = 9.2 \pm 3.0$ ms, 52.5 ± 23.5%, $\tau2 = 40.6 \pm 21.2$ ms; p = 0.91, p = 0.65 and p = 1.00 respectively, n = 9 cells). These results confirm the presence of a glycinergic component at the iNC-Golgi cell synapses albeit with large variability in its magnitude (range: 0–63%; *Figure 3C2*, left). Subsequent application of a GABA$_A$-receptor antagonist (gabazine, 2 µM), almost completely blocked the response, decreasing the amplitude of the IPSC by 96.5 ± 2.9% (p = 0.0078, n = 8 cells; *Figure 3C1,2*). These results confirm the mixed GABAergic-glycinergic nature of the iNC axons.

To characterize the functional effect of the iNC-originating inhibitory currents on Golgi cells' firing, we recorded Golgi cells in the current-clamp mode in acute slices obtained from GAD-cre mice.

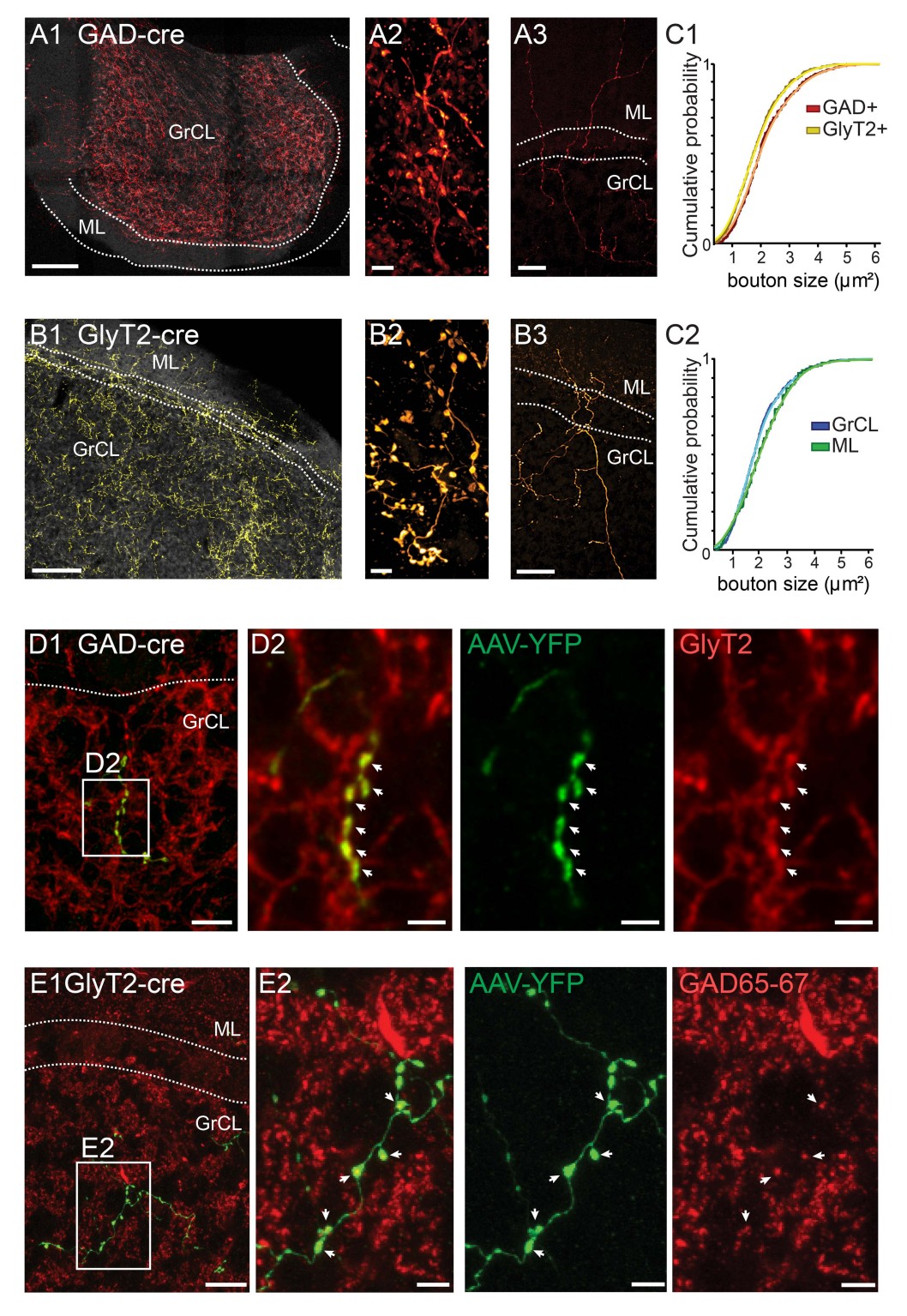

**Figure 2**. iNC axons are found in cerebellar granule cell and molecular layers and contain GAD65-67 and GlyT2.
(**A–B**) Confocal composite images of sections through the flocculus in GAD-cre (**A1**) and posterior vermis in GlyT2-
cre (**B1**) mice, showing dense iNC axons in the GrCl as well as sparse axons in the ML (arrows). 40× composite tiles.
Large axonal swellings from both GABAergic (**A2**) and glycinergic (**B2**) axons are found in the GrCL. Both
GABAergic (**A3**) and glycinergic (**B3**) iNC axons occasionally rise into the lower ML. (**C**) Comparison of iNC axonal
bouton sizes between the GABAergic and glycinergic axons (**C1**) and between the boutons in the GrCL and ML
*Figure 2. continued on next page*

*Figure 2. Continued*

(**C2**) shows nearly identical distributions. (**D**) Merged confocal image (Z-projection thickness: 12.2 μm) showing iNC axons in GAD-cre mice injected with AAV-flox-EYFP (green) are co-stained for GlyT2 (red) (**D1**). Higher magnification of axonal swellings (arrows) co-stained for EYFP and GlyT2 (**D2**). (**E**) iNC boutons (green) transfected with AAV-flox-EYFP in GlyT2-cre mice are stained for GAD65-67 (red, **E1**, Z-projection thickness: 8.2 μm). (**E2**) Higher magnification of iNC axon (arrows) co-stained for EYFP and GAD65-77 (Z-projection thickness: 2.4 μm). Abbreviations: GrCL, granule cell layer; ML, molecular layer; PNL, Purkinje neuron layer; WM, white matter; n.s., non-significant. Scale bars: **A1** and **B1**: 100 μm; **A2** and **B2**: 5 μm; **A3** and **B3**: 50 μm. **D1**: 20 μm. **D2–4**: 5 μm. **E1**a-e: 10 μm; **E2**a-e: 2 μm.

Whole-field light stimulation of the iNC axons had a significant effect on the spiking in 24 out of 86 recorded Golgi cells (27%; *Figure 3D–G*). A single 5-ms light pulse elicited clear inhibitory responses in most of cases, involving a hyperpolarizing post-synaptic potential (n = 12 cells; IPSP amplitude 2.1 ± 1.5 mV; *Figure 3D*) and/or prolongation of the inter-spike interval (ISI) during which the stimulation occurred (n = 15 cells; 60 ± 30% increase; on average, 223 ± 123 ms ISI increased to 355 ± 286 ms; cells; p = 0.0001, paired t-test; *Figure 3E1,E2*). This inhibition of spiking was more pronounced when iNC fibers were activated with a train of 4–5 light pulses at 50 Hz (pulse duration 10 ms), eliciting a longer spike delay (71 ± 44% increase; n = 16 cells, p < 0.05; *Figure 3F*). In some of the recorded Golgi cells, trains of light pulses elicited a time locking of intrinsic spikes (n = 7, *Figure 3—figure supplement 1*) without clear inhibitory effect, suggesting a network effect mediated through the gap junctions among Golgi cells (*Dugué et al., 2009*; *Vervaeke et al., 2010*). Taken together, these results demonstrate that the iNC pathway inhibits spiking in Golgi cells.

## iNC fibers inhibit a subpopulation of Golgi cells with characteristic electrophysiological properties

The rate of success in finding responsive Golgi cells was rather low (24% and 27% of all recorded Golgi cells in GlyT2-cre and GAD-cre mice, respectively) suggesting that iNC axons might preferentially or exclusively inhibit a certain subpopulation of Golgi cells. In the following, we will present both electrophysiological and immunochemical evidence supporting this possibility.

While analyzing our current-clamp recordings in GAD-cre mice, we noted considerable variability in Golgi cells' properties, with their spontaneous spiking showing the most striking difference (*Figure 4A*). While 64 out of 85 recorded Golgi cells fired spontaneously at low rates (mean frequency: 9.0 ± 6.5 Hz), the other 21 Golgi cells were quiescent and had a resting membrane potential negative to the spiking threshold (resting potential −55 ± 2.4 mV, spike threshold −43 ± 1 mV, n = 13 cells). When evaluating iNC effects in Golgi cells, it became obvious that only the spontaneously active Golgi cells ('s-Golgi cells') were responsive to iNC stimulation. Specifically, 24 out of 64 (37.5%) s-Golgi cells were inhibited by iNC activation (*Figure 4A–B*; blue), whereas none of the 21 not-spontaneously spiking Golgi cells ('ns-Golgi cells'; green) were affected by the stimulation. Repeating these stimulations during depolarizing current injections that drove the ns-Golgi cells to continuous spiking also failed to reveal iNC effect (*Figure 4A*, top right). These findings suggested that iNC fibers inhibit preferentially s-Golgi cells. It should be noted that our virus injections were unlikely to result in transfection of the entire iNC population, hence the observed fraction of inhibited s-Golgi cells is bound to be underestimated.

The s-Golgi cells differ from ns-Golgi cells also in action potential (AP) shape (*Figure 4D*) recorded during steady-state firing. The observation that the AP waveform (composed of AP and after-hyperpolarization [AHP]) was shorter in s-Golgi cells compared to ns-Golgi cells (*Figure 4E*; see *Table 1*) led us to seek for other distinguishing electrophysiological features. Compared to the s-Golgi cells, the ns-Golgi cells showed significantly larger variability in all of the AP shape measurements, although no significant differences were found in their average values (see *Table 1*). Also, no differences were found in population averages of the input–output relationship of the two Golgi cell groups, as evidenced by nearly identical current-to-firing frequency (I-F) curves (*Figure 4F*; *Table 1*). ns- and s-Golgi cells did not differ as a population in their estimated capacitance ($C_m$), nor in their input resistance, but analysis of their variance showed clear differences between the groups (*Figure 4G*, compare the significance values obtained with Wilcoxon and F-tests in *Table 1*). The population variability became most visible when comparing the steady-state

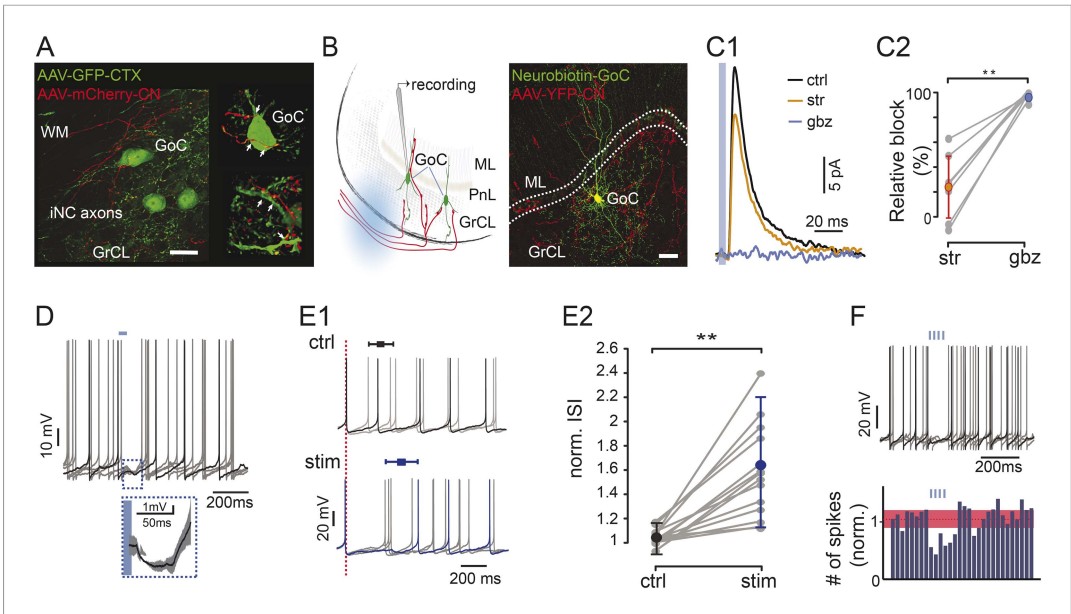

**Figure 3**. Optogenetic stimulation of iNC axons in cerebellar slices inhibits Golgi cells' spiking. (**A**) Confocal image (left) and reconstruction (right) of GrCL in GAD-cre mouse injected with AAV2-flox-ChR2-mCherry to the CN and AAV-GFP to the cerebellar cortex. iNC fibers (red) branch in the GrCL and form axonal swellings (arrows) on GoCs (green). (**B**) Left: schematic drawing of the in vitro experimental arrangement. GoCs were recorded in GlyT2-cre or GAD-cre animals where iNC axons express both ChR2 and a fluorescent marker (YFP in GlyT2-cre, mCherry in GAD-cre). Right: GoC patched and filled with Neurobiotin (green) in GlyT2-Cre mice surrounded by transfected axons (red) (Z-projection thickness: 36.4 μm; Sagittal view). (**C**) Optogenetic stimulation of the iNC fibers. (**C1**) An example of averaged IPSCs (n = 30) recorded in Golgi cell induced by 5-ms illumination (indicated by blue line), blocked by successive bath application of 300 nM strychnine (str; orange), and 2 μM gabazine (gbz, blue). (**C2**) Summary plot of the percentage of inhibitory current blocked by strychnine and gabazine (n = 9; p = 0.0039). (**D**) Example voltage traces from a recorded Golgi cell with a 50-ms single light pulse in GAD-Cre mice. The averaged IPSP response (± STD) of all 6 traces is magnified in the inset. (**E**) iNC activation delays spike generation in GoCs. (**E1**) The traces recorded without (top; ctrl; black) and with (bottom; stim; blue) light stimulation are aligned either on the first spike in the sweep (top) or on the spike preceding the stimulus (bottom; red, dashed line) to emphasize the increased ISI in response to iNC stimulation. Average inhibition delay (± STD) for the example cell is marked above traces in Box-and-whiskers symbols. (**E2**) Comparison of the average ISI without and with light pulse (± STD) normalized to the average ISI. (p = 0.0001; n = 15). (**F**) Example voltage traces from a recorded GoC during train of light pulses stimulation of iNC axons showing no spikes occurring during the illumination (upper panel). Peri-stimulus time histograms (PSTH) of the GoCs (n = 16, 100-ms bin) shows a decrease in the number of spikes after train pulse stimulation. Baseline average marked in dashed red line and STD values in red area (all cells normalized to baseline frequency; lower panel). Scale bars: **A**: 20 μm, **B**: 50 μm. 'Asterisks' indicate statistical significance. Abbreviations: WM, white matter; GrCL, granule cell layer; GoC, Golgi cell; ML, molecular layer; ISI, inter-spike interval. See also *Figure 3—figure supplement 1*.

The following figure supplement is available for figure 3:

**Figure supplement 1**. iNC activation modulate spike times in a fraction of non-responsive s-Golgi cells.

frequency accommodation (*Figure 4H*, left, compare the significance values obtained with Wilcoxon and F-tests in *Table 1*): the s-Golgi cells accommodated very uniformly to roughly half of the initial firing frequency, while ns-Golgi cells showed either no adaptation (evidenced by steady-state accommodation values around 90% of control) or adapted even more than the s-Golgi cells (to 35% of control; compare the widths of blue and green bars in *Figure 4H*). The large variability of ns-Golgi cells in frequency accommodation, AHP time, AP half width, $C_m$, input resistance, and AP shape suggest that the ns-Golgi cells form a heterogeneous group of cells consisting of several functionally distinct subpopulations.

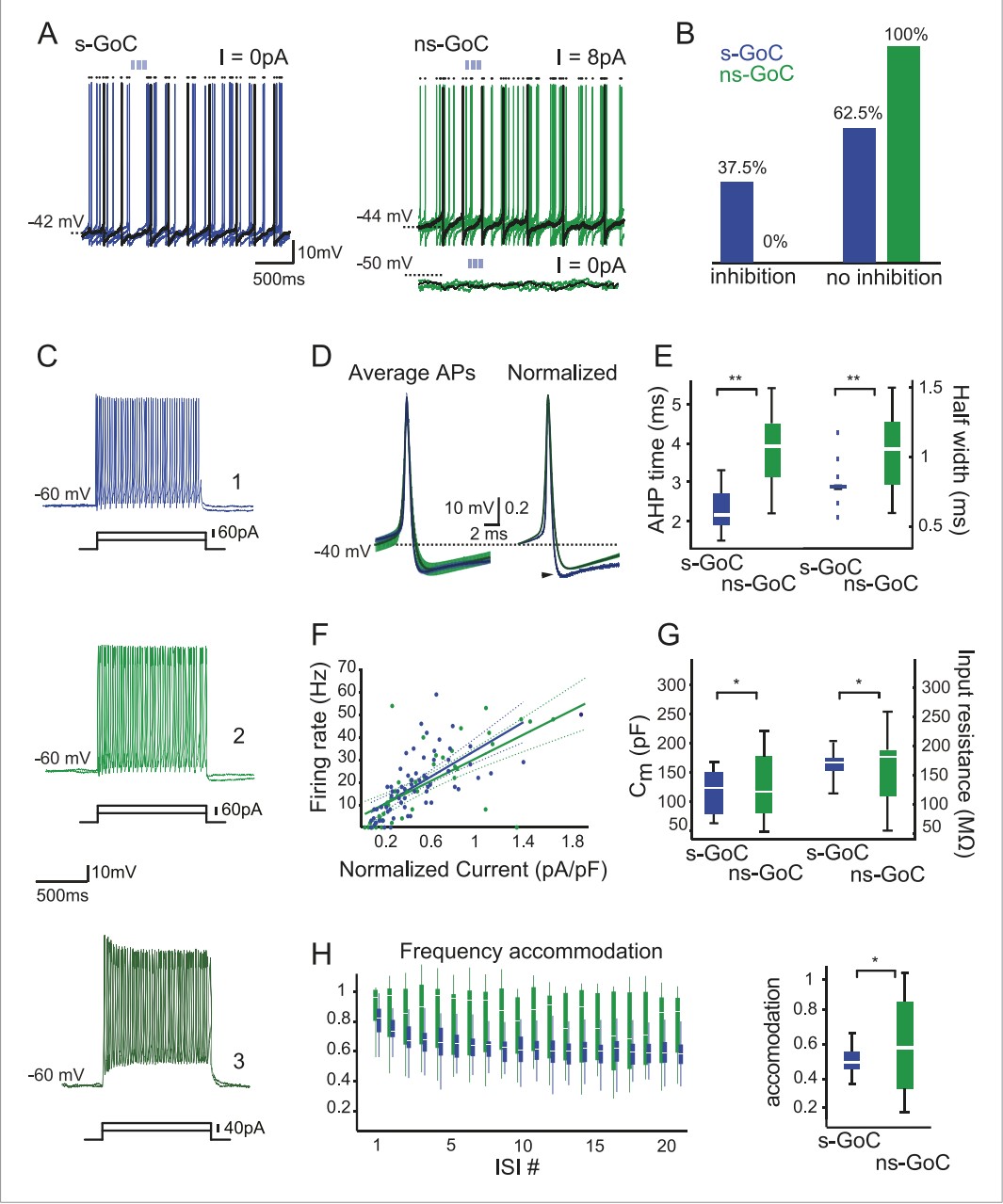

**Figure 4**. Golgi cell subtypes differ in their sensitivity to iNC input and have different intrinsic properties. (**A**) Left: a spontaneously active GoC (s-GoC) is inhibited by optogenetic iNC activation (light-blue bars above traces). Six superimposed traces with no holding current. (**A**) Right: a not-spontaneously spiking GoC (ns-GoCs) shows no response to iNC activation. Bottom trace: without depolarizing current injection; top trace: with +8 pA current injection to evoke spiking. Black dots above traces in both panels mark spike times. (**B**) Percentage and numbers of s-GoCs (blue) and ns-GoCs (green) that are inhibited by iNC axons (left) and those that are not (right). (**C**) Example traces of three GoCs' responses to positive current steps. Cell 1, blue: s-GoC ($C_m$ = 173.5 pF); Cell 2, dark green: a large ns-GoC ($C_m$ = 125.3 pF); Cell 3, light green: small ns-GoC ($C_m$ = 73.0 pF). (**D**) AP waveforms differ between s-GoCs and ns-GoCs. Left: superimposed, grand average AP shapes (± STD) obtained from s-GoCs (n = 31 cells, blue) and ns-GoCs (n = 13 cells, green) during steady-state firing. Right: APs are peak-normalized (± SEM). s-GoCs show faster spike repolarization as well as faster after-hyperpolarization (arrowhead). (**E**) Comparison of AP parameters shows that s-GoCs (n = 31 cells) spikes are faster than those in ns-GoCs (n = 13 cells). (**F**) Current-to-firing frequency (IF) relationship of s-GoCs (blue, n = 23) and ns-GoCs (green, n = 11) is not significantly different. The solid and dashed lines show fitted single polynomials and the confidence intervals, respectively. The current injection values are normalized to the estimated $C_m$ of the cells. (**G**) $C_m$ and input resistance of s-GoCs (blue; n = 23) and ns-

*Figure 4. Continued*

GoCs (green; n = 11). (**H**) Comparison of instantaneous firing frequency accommodation during a depolarizing step. Left: box-plot chart showing the development of frequency accommodation in s-GoCs (blue) and ns-GoCs (green). For visual clarity, the s-GoC bars are slightly shifted to the right in respect to the ns-GoC. Right: box plots of the steady-state accommodation among s-GoCs (blue) and ns-GoCs (green) show that s-GoCs have a smaller range of accommodation than ns-GoCs (t-test, p = 0.008). 'Asterisks' denote statistical significance. Abbreviations: s-GoC, spontaneously spiking Golgi cell; ns-GoC, not-spontaneously spiking Golgi cell, AP, action potential; AHP, after-hyperpolarization; acc val, accommodation value.

## iNC axons contact Golgi cells with a specific immunohistochemical profile

The results described above suggest that iNC fibers specifically target a subpopulation of spontaneously active Golgi cells with uniform electrophysiological properties. It is well established that Golgi cells are neurochemically heterogeneous (*Ottersen et al., 1988*; *Simat et al., 2007*; *Pietrajtis and Dieudonné, 2013*). While the majority of Golgi cells express the glycine transporter GlyT2, with most of them being of mixed GABA-glycinergic phenotype, about 15–20 % of Golgi cells are purely GABAergic (*Ottersen et al., 1988*; *Simat et al., 2007*). GlyT2-expressing Golgi cells have previously been reported to be intrinsically silent (*Dugué et al., 2009*). We, thus, hypothesized that the iNC-inhibited Golgi cells, all of which fire spontaneously, may correspond to the purely GABAergic Golgi cells. As all of the Golgi cells that express the calcium-binding protein, neurogranin are also GABAergic (*Simat et al., 2007*), we used neurogranin in conjunction with GlyT2-eGFP expression to differentiate pure GABAergic Golgi cells from the mixed GABA-glycinergic and pure glycinergic Golgi cell populations (*Simat et al., 2007*; *Pietrajtis and Dieudonné, 2013*).

We designed a strategy to identify the subtypes of Golgi cells targeted by iNC axons. GlyT2-eGFP transgenic mice (*Zeilhofer et al., 2005*), in which both mixed GABA-glycinergic and pure glycinergic Golgi cells are labeled with eGFP, were mated with GlyT2-cre animals. The CN of the offspring carrying both transgenes were injected with a floxed AAV expressing the red fluorescent protein tdTomato. Cerebellar cortical sections from these mice were then stained for neurogranin to differentiate between the Golgi cells subtypes. iNC axons were easily identified by their co-expression of eGFP and tdTomato and were found to preferentially contact cell bodies and dendrites of Golgi cells that were intensely stained for neurogranin (*Figure 5A–B*). This selective targeting of neurogranin-positive cells by iNC fibers extended to the ML (indicated by arrowheads in *Figure 5C*), as iNC fibers were seen to climb along the apical dendritic shafts of neurogranin-positive Golgi cells to the ML. Similarly, in GAD-cre animals, iNC fibers transfected with YFP and co-stained for GAD65-67 were found to impinge on the dendrites and cell bodies of Golgi cells strongly expressing neurogranin (*Figure 5D*, arrowheads).

In most cases, the innervated Golgi cells were devoid of eGFP staining, suggesting that they are non-glycinergic Golgi cells. However, a few of the iNC-contacted Golgi cells exhibited a low level of eGFP staining in their somata (examples are indicated by asterisks in *Figure 5A,B*). To distinguish between the eGFP positive and negative Golgi cell subpopulations in a more objective manner, we quantified the normalized mean GlyT2-eGFP and neurogranin staining intensities at the somata of Golgi cells (n = 317 cells, n = 13 stacks n = 4 animals; see 'Materials and methods'). The 2D distribution of the two staining intensities was separated into two populations based on k-means statistical clustering (*Figure 5E*; green population: n = 238 Golgi cells (75%); blue population: n = 79 Golgi cells (25%); see 'Materials and methods'). Golgi cells in the two groups differed mainly in their eGFP staining, as illustrated by the bimodal distribution of the mean eGFP intensities (*Figure 5F*). The normalized neurogranin intensities were approximately 50% higher in the low-eGFP population (Wilcoxon test p < 0.00001), even though distributions overlapped extensively (*Figure 5G*). We, thus, consider that the blue population of *Figure 5* constituting of Golgi cells expressing neurogranin and none or low levels of GlyT2-eGFP corresponds to the population of Golgi cells releasing principally or exclusively GABA (*Aubrey et al., 2007*). For the sake of brevity, we will refer to these cells as 'GABAergic Golgi cells' in the following.

We counted the iNC appositions found on the somata and large proximal dendrites of each Golgi cell (see 'Materials and methods'; color-coded in *Figure 5H*) as a measure of connection strength.

**Table 1.** Summary of s-Golgi and ns-Golgi cells spiking parameters

|  | s-Golgi | ns-Golgi | N (s) | N (ns) | p-value (Wilcoxon) | p-value (F-test) |
|---|---|---|---|---|---|---|
| AP half-width (ms) | $0.8 \pm 0.2$ | $1.2 \pm 0.4$ | 33 | 11 | 0.01 | 0.001 |
| AP threshold (mV) | $38.9 \pm 6.04$ | $-36.4 \pm 9.4$ | 33 | 11 | 0.8 | 0.1 |
| AP amplitude (mV) | $51.2 \pm 9$ | $43.5 \pm 10.7$ | 33 | 11 | 0.06 | 0.4 |
| AP peak voltage (mV) | $22.4 \pm 8.7$ | $16.8 \pm 13.03$ | 33 | 11 | 0.1 | 0.17 |
| AHP min voltage (mV) | $-51.9 \pm 4.5$ | $-51.5 \pm 8.8$ | 33 | 11 | 0.8 | 0.01 |
| AHP time (ms) | $2.1 \pm 0.8$ | $4.0 \pm 0.5$ | 33 | 11 | 0.009 | 0.03 |
| AHP amplitude (mV) | $24.3 \pm 4.2$ | $23.1 \pm 8.4$ | 33 | 11 | 0.9 | 0.003 |
| I-F slope ($r^2$) | 0.5 | 0.4 | 23 | 11 | – | 0.3 (cov-analysis) |
| Freq. Acc. (%) | $53 \pm 8\%$ | $59 \pm 29\%$ | 23 | 11 | 0.3 | $1.1 \times 10^{-6}$ |
| $C_m$ (pF) | $127.5 \pm 48.3$ | $118.9 \pm 78.3$ | 23 | 11 | 0.052 | 0.0018 |
| $R_m$ (M$\Omega$) | $185.5 \pm 43.2$ | $161 \pm 75.5$ | 23 | 11 | 0.054 | $8.3 \times 10^{-5}$ |

Most of the Golgi cells that were contacted by at least one iNC bouton were GABAergic Golgi cells (80%, n = 32 out of the 40 'iNC-contacted' cells; *Figure 5H,I*). In contrast to the glycinergic Golgi cells that were only rarely apposed to iNC boutons (3%, n = 8 out of 238 all Golgi cells; on average 1.88 ± 1.12 appositions per cell; max = 4 appositions; 15 appositions found overall; *Figure 5I*, blue dots), 41% of the GABAergic Golgi cells were contacted by on average 7.65 ± 4.69 appositions (n = 32 out of 79 Golgi cells, max = 20 appositions; 245 appositions found overall; *Figure 5I*, blue dots). These results demonstrate that iNC fibers contact almost exclusively the GABAergic Golgi cell population, as summarized in the schematic drawing of *Figure 5J* (97% of iNC terminals contact the GABAergic Golgi cell population). To the best of our knowledge, this is the first evidence for differential connectivity among subpopulations of Golgi cells.

## Bursting activity in iNC neurons and their inhibitory effect on Golgi cell activity in vivo

The impact of a neuronal pathway depends on properties of transmission at its synapses as well as the firing pattern of its neurons. In the case of the iNC pathway, repetitive stimulation of the axons evoked stronger inhibition of Golgi cells (*Figure 3G*). To investigate the physiological relevance of such burst activation of iNC axons, we examined the responses of iNC neurons to optogenetic stimulation. First, using acute slices from GlyT2-cre mice transfected as above, we performed extracellular recordings from iNC neurons, identified by their YFP fluorescence. iNC neurons were silent (n = 11 cells) in contrast with the other cell types in the CN (*Uusisaari and Knöpfel, 2010, 2012*). Optogenetic excitation of iNC neurons by short light pulses (1 ms) evoked high-frequency bursts of spikes (*Figure 6A1*). Increasing the illumination power resulted in an increased number of spikes and in mean burst frequency, which saturated around 450 Hz for a light power of 1–2 mW/mm (*Figure 6A2*; n = 11 cells).

To further characterize intrinsic bursting, we performed whole-cell current-clamp recordings from iNC neurons (identified by their fluorescence and lack of spontaneous activity) in slices from transfected GAD-cre mice (*Figure 6B1*, n = 3 cells). At saturating illumination intensity (1.3 mW/mm$^2$), stereotypical high-frequency bursts of spikes were evoked with short light durations (10 ms), resembling the extracellular recordings in the GlyT2-cre slices. These bursts were riding on a depolarized plateau, which outlasted the illumination period, and were often followed by a prolonged depolarized after-potential and low-frequency firing. Increasing the duration of the light pulse extended the burst duration without affecting the intra-burst frequency (*Figure 6B2*). These results indicate that high-frequency bursting of APs could constitute the main firing mode of iNC neurons in response to excitatory synaptic inputs.

To investigate the physiological significance of the iNC pathway in an intact cerebellum, we implanted an optical fiber in the CN of virally transfected GlyT2-cre mice to optically activate the iNC neurons, while recording Golgi cell activity (*Figure 6C1*). Based on our in vitro calibration (*Figure 6A,B*), single 25-ms long light pulses are expected to evoke short bursts of firing in the iNC neurons. This

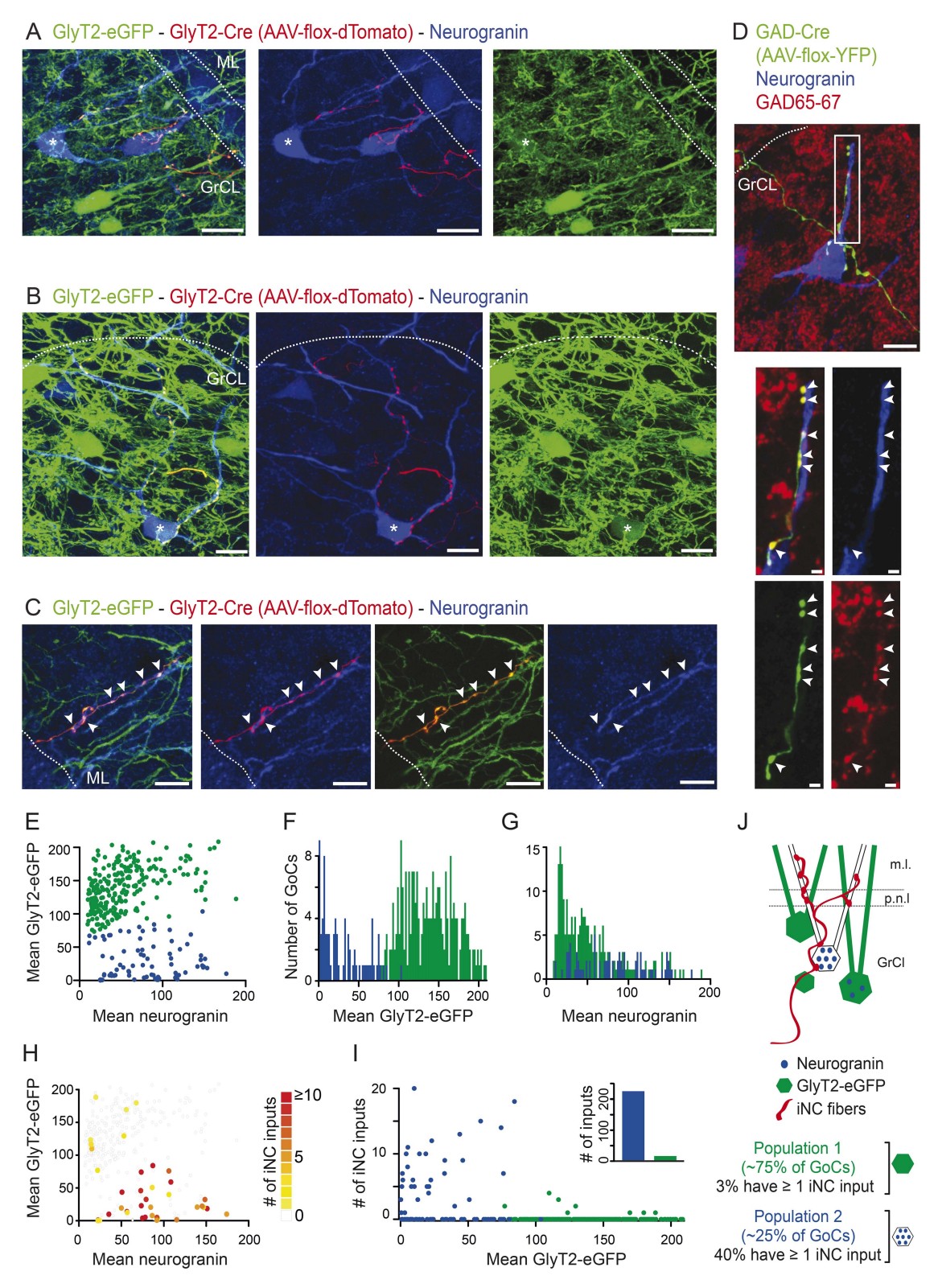

**Figure 5**. iNC fibers contact preferentially a neurochemically distinct subtype of Golgi cells expressing neurogranin. (**A–C**) iNC fibers were transfected with AAV-flox-tdTomato in GlyT2-Cre X GlyT2-eGFP mice. iNC fibers were identified in the cerebellar cortex by their co-labeling for both GFP (green) and tdTomato (red). (**A–B**) In the GrCL, iNC fibers contact somata and proximal dendrites of neurogranin-expressing (blue) GoCs, either devoid of GFP staining or exhibiting a faint GFP staining at their somata (indicated by 'asterisks'). (**C**) In the ML, GFP-positive / tdTomato-positive iNC fibers (arrowheads)

*Figure 5. continued on next page*

Ankri *et al.* eLife 2015;4:e06262. DOI: 10.7554/eLife.06262

*Figure 5. Continued*

were also seen apposed to GoC apical dendrites stained for neurogranin (blue) and virtually devoid of GFP staining (green). (**D**) iNC fibers, transfected with AAV-flox-YFP virus (green) in GAD-cre mice, contacted neurogranin-positive (blue) GoCs. iNC varicosities (arrowheads) are co-stained for GAD65-67 (red). (**E**) Plot of the mean GlyT2-eGFP intensities over mean neurogranin intensities allows statistical discrimination (k-means 2D) between two GoCs populations. A first population (blue) was distinguished from the second population (green) by its none-to-low levels of GlyT2-eGFP staining, as seen with the bimodal distribution of mean GFP intensities (**F**), while the mean neurogranin intensities were less discriminative (**G**). According to the color-coded number of iNC inputs received by each GoC (**H**), most of the 'iNC-contacted' GoCs were found in the neurogranin-positive/GlyT2-eGFP negative GoC population (blue) (**I**). (**J**) Schematic drawing of the percentages obtained for each GoC subtypes. Z-projection thickness: **A**: 34 μm; **B**: 18.4 μm; **C**: 16.3 μm; **D**: 30 μm; **D** close up: 2.4 μm. Scale bar: **A**: 20 μm, **B–D**: 10 μm, **D** close up: 2 μm. Abbreviation: GoC, Golgi cell.

illumination protocol suppressed spiking in 18 out of 86 recorded Golgi cells (21%, *Figure 6C2*, left). The rest of the Golgi cells (79%, *Figure 6C2*, middle) as well as PNs (n = 50 cells, *Figure 6C2*, right) did not show any significant modulation of the spiking frequency following illumination. The time course of the inhibition in the responsive Golgi cells was variable (duration: 23.4 ± 11.7 ms; onset latency: 14.5 ± 7.2 ms; peak latency: 25.4 ± 14.1 ms; n = 18, *Figure 6D2*) as exemplified with colored traces from individual cells in *Figure 6D1*. The variability of the inhibitory effect can be explained by the variability in iNC spike-burst duration that depends on the distance from the optic fiber and thereby stimulation light intensity (*Figure 6A*). Regardless of this variability, Golgi cells' firing was robustly suppressed (frequency decreased to 1.58 ± 1.46 Hz from a baseline of 10.9 ± 3.9 Hz, n = 18 cells, *Figure 6D3*). Interestingly, the average firing rate (FR) of responsive Golgi cells was significantly higher than the average FR of non-responsive Golgi cells (10.5 ± 3.5 Hz, n = 18 cells vs 8.2 ± 4.2 Hz, n = 68 cells, respectively; Wilcoxon test: p = 0.036; *Figure 6D4*). While we cannot make a direct link between the lower FR of non-responsive Golgi cells in vivo and the quiescence of ns-Golgi cells in vitro, these results are supporting the notion that the iNC pathway is targeting a distinct group of Golgi cells.

Overall, our results provide the first functional evidence for an iNC pathway suppressing GABAergic Golgi cell spiking. This pathway likely modulates the inhibitory control of GrCs and thereby gating of sensori-motor inputs into the cerebellar cortex.

## Discussion

In the present work, we reveal an iNC pathway in the cerebellum. This projection is formed by mixed GABA-glycinergic neurons of the CN and targets the GABAergic Golgi cells in the cerebellar cortex.

### The iNC pathway and identity of the iNC cells

Anatomical demonstrations of nucleo-cortical pathways have appeared in literature already decades ago (*Tolbert et al., 1976*; *Gould and Graybiel, 1976*; *Dietrichs and Walberg, 1979*; *Hámori et al., 1980*; *Buisseret-Delmas, 1988*; *Batini et al., 1992*; reviewed in *Haines and Manto, 2009*; *Houck and Person, 2013*). These classical studies, often ignorant of the afferents' neurotransmitter type, described a range of nucleo-cortical axonal morphologies including rosette-like and simple terminals (*Hámori et al., 1980*; *Tolbert et al., 1980*). It was only later established that both glutamatergic (*Tolbert et al., 1980*; *Payne, 1983*; *Batini et al., 1992*; *Houck and Person, 2015*) and GABAergic (*Hámori and Takács, 1988*; *Batini et al., 1989*, *1992*; *Houck and Person, 2015*) CN neurons project to the cortex. Here, using targeted viral transfection and labeling, we demonstrate that the iNC axons originate from a population of mixed GABA-glycinergic CN neurons. The iNC axon terminals were simple in their morphology, and rosette-like structures were never observed. Thus, the GABAergic rosette-like terminals found in GrCL glomeruli described in earlier works (*Chan-Palay et al., 1979*; *Hámori and Takács, 1988*) must arise from extracerebellar sources. The morphology and spread of the iNC axons as well as the axonal bouton size was also different from both the Golgi and Lugaro axons (*Dieudonné, 1998*; *Dumoulin et al., 2001*).

Our study discards the suggestion that iNC axons would emerge as collaterals of GABAergic NO neurons (*Figure 1*; *Tolbert et al., 1978*; *Haines, 1988*). The neurons transfected in the GlyT2-cre animals do not include NO cells, as evidenced by the lack of labeling in the IO (*Husson et al., 2014*; see also *De Zeeuw et al., 1994*) and the clear difference in cell body size between GlyT2-cre and NO neurons (*Figure 1B–D*). While viral transfection protocols used in the GAD-cre mice also transfect NO cells (*Lefler et al., 2014*; *Figure 1A3*), all the fibers found in the cortex were GlyT2 immunopositive,

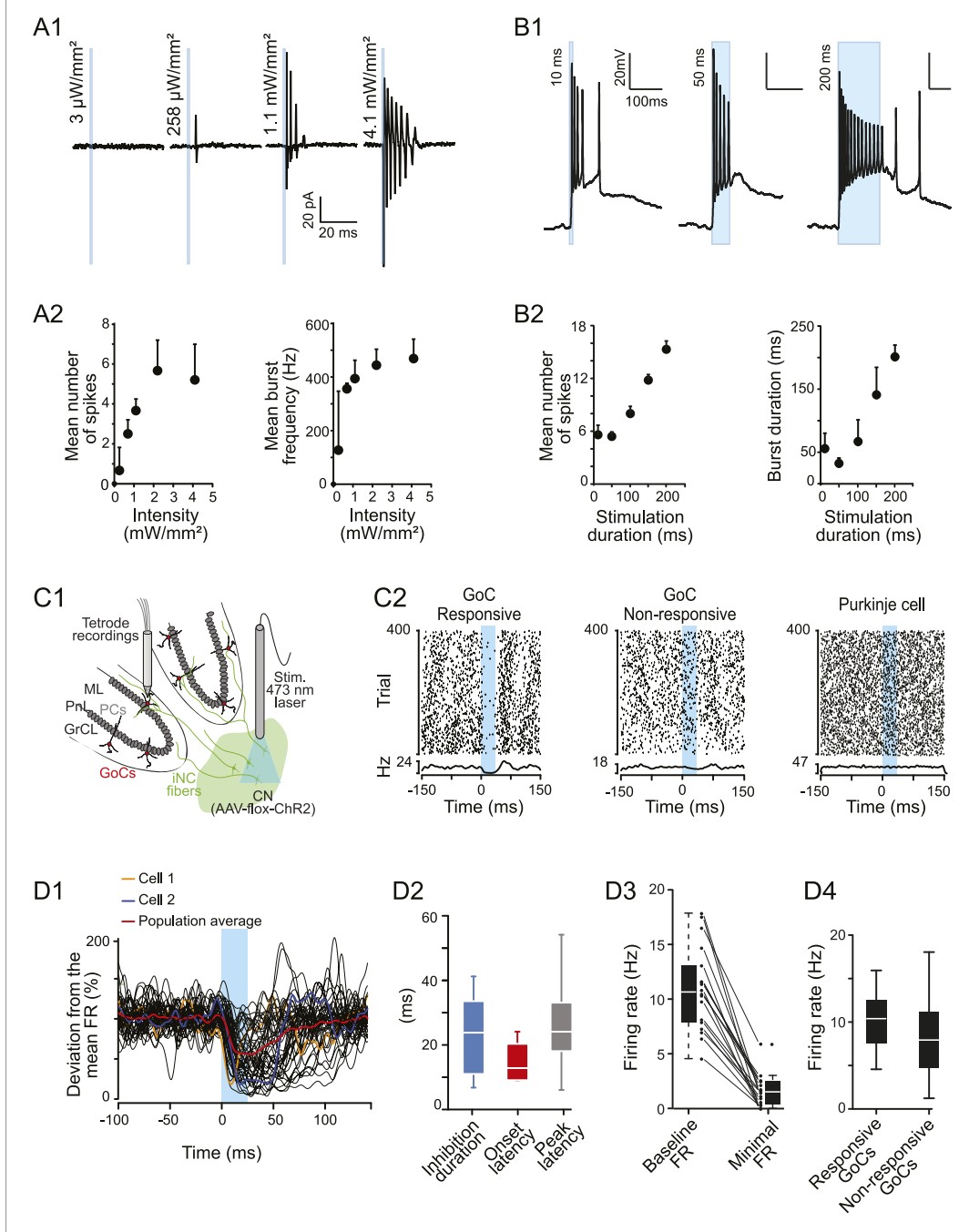

**Figure 6**. iNC neurons exhibited a burst firing phenotype and their optogenetic stimulation has inhibitory effects on Golgi cells firing in vivo. (**A1**) Extracellular recordings of iNC neurons in GlyT2-Cre mice transfected with ChR2 virus during increasing intensity of stimulation (1-ms duration pulse; blue bars). (**A2**) iNC neurons exhibit a burst firing phenotype with increase of mean number of spikes per burst (left) and mean burst frequency (right) when increasing illumination intensity. (**B1**) Whole-cell current-clamp recording of GAD-Cre iNC neurons transfected with ChR2 during 10-, 50-, or 200-ms long light pulse (blue bars; light intensity: 1.3 mW/mm$^2$) has burst firing phenotype. (**B2**) GAD-Cre transfected iNC neurons show increase of their mean number of spikes per burst (left) and mean burst duration (right) with increasing illumination duration. (**C1**) Schematic drawing of the experimental system for in vivo recordings: extracellular recordings of GoCs during 25-ms pulse illumination of the CN in anesthetized GlyT2-Cre mice injected with ChR2 in the CN. (**C2**) Raster plots of two GoCs recorded at the same time and of one PN recorded in the same area, with their corresponding PSTH. Light pulse start at 0 ms. (**D1**) All superimposed smoothed PSTHs of responsive GoCs (18 out of 86 recorded GoCs), normalized to their mean firing rate (FR), with

*Figure 6. continued on next page*

*Figure 6. Continued*

the population average trace (red). Two individual traces are highlighted (orange and blue), illustrating the high variability of the inhibition period parameters. Smoothed PSTHs are obtained by convolving 1-ms time bin PSTHs with a Gaussian kernel with 3 ms standard deviation. (**D2**) Characterizing parameters of the responses. (**D3**) Light stimulation of iNC neurons decreased the firing rate. (**D4**) Comparison of responsive and non-responsive GoCs firing rates. Abbreviations: GoC, Golgi cell.

demonstrating that only those GABAergic CN cells that also express GlyT2 project to the cortex. Also, as the purely glycinergic neurons of the medial CN nucleus projecting to the vestibular nuclei (*Bagnall et al., 2009*) are not found in the main targets of viral transfections in the present study (interpositus and lateral CN), they are unlikely to be the source of the iNC axons.

The nucleo-cortical axons in both GlyT2-cre and GAD-cre models were very similar in shape and function and co-stained for GAD and GlyT2, respectively (*Figure 2*). The small differences observed are likely to originate from variability in fixation procedures and wavelength dependence of optical resolution. Therefore, iNC fibers undoubtedly represent the axons of a single mixed CN neuron type. The density of nucleo-cortical fibers in the GAD-cre model was somewhat higher than in the GlyT2-cre model (compare panels 2A1 and 2B1), most likely due to the mosaic expression of cre in only 50% of mixed neurons in the GlyT2-cre mice (*Husson et al., 2014*) as well as the stronger expression levels obtained with the AAV9 serotype virus used in the GAD-cre model.

The iNC neurons described in the present work show clear morphological (*Figure 1B2*) and electrophysiological (*Figure 6*) resemblances to the CN glycinergic neurons described in two recent studies as spontaneously inactive, mixed GABA-glycinergic neurons (compare the present results with Figure 1 in *Uusisaari and Knöpfel, 2010* and Figure 7Ab in *Husson et al., 2014*). Thus, we conclude that the iNC neurons, the 'Gly-I' neurons, and the mixed GABA-glycinergic neurons are the same cells. While these neurons also have local axon collaterals within in the CN (*Husson et al., 2014*), their projection to the cortex is their most distinguishing feature. Thus, we propose that they should be referred to as 'iNC' neurons.

## Diversity of Golgi cells and their inhibitory control

Golgi cells have previously been shown to receive inhibitory synapses from both Lugaro and other Golgi cells in the cerebellar cortex. We demonstrate here that single iNC projection axons form numerous terminal swellings on the somata and dendrites of Golgi cells (*Figure 2*), somewhat reminiscent of the climbing fiber articulation on Purkinje cells. Specific optogenetic stimulation of the iNC axons evokes IPSCs mediated both by GABA$_A$ and glycine receptors (*Figure 3*), in line with the immunohistochemical evidence that iNC terminals contain both GABA and glycine (*Figure 2D–E*). The average synaptic conductance at the iNC synapses (estimated to be 1.9 and 8.5 nS in physiological and symmetrical chloride, respectively) is about six times the conductance reported at unitary Golgi–Golgi synapses (0.33 nS in physiological chloride; *Hull and Regehr, 2011*) and similar to the conductance at Lugaro to Golgi cell synapses in the juvenile animal (*Dumoulin et al., 2001*). As our spatially restricted light stimulation likely activated a single or only a few iNC axons, the evoked IPSCs likely represent unitary responses through the multiple contacts made by single axons on Golgi cells.

Most of the current physiological work on Golgi cells tends to assume a homogeneous neuronal population while simultaneously using various, partly contradictory, identification criteria to target them for experiments (*Schulman and Bloom, 1981*; *Holtzman et al., 2006*; *Xu and Edgley, 2008*; *Hull and Regehr, 2011*; *Hull et al., 2013*). However, accumulating evidence indicates that Golgi cells can be divided into different groups based on their neurotransmitter content (GABA, glycine or both; *Ottersen et al., 1988*) as well as specific molecular marker expression (*Simat et al., 2007*; *Dugué et al., 2009*; *Pietrajtis and Dieudonné, 2013*). By demonstrating that the Golgi cells contacted by the iNC axons are characterized by none or low level of GlyT2-eGFP staining as well as high-neurogranin expression (*Figure 5*; compare with 'type 4' Golgi cells in *Simat et al., 2007*), we present the first evidence that neurochemical subtypes of Golgi cells may participate in specific microcircuits.

Our work further supports the functional specialization of purely GABAergic Golgi cells by showing that they can be distinguished from other Golgi cells based on their electrophysiological properties (*Figure 4*). A previous work showed that glycinergic (GlyT2-eGFP positive) Golgi cells are not spontaneously active in vitro (*Dugué et al., 2009*). Here, we show that spontaneously spiking Golgi

cells (s-Golgi cells), unlike the not-spontaneously spiking Golgi cells (ns-Golgi cells), receive functional synaptic contacts from the iNC fibers (*Figure 3*). s-Golgi cells had relatively uniform properties (*Figure 4*), confirming their identification as a distinctive functional group. In contrast, ns-Golgi cells varied in all of the examined features (*Figure 4* and *Table 1*) suggesting that this population may be further divided into several functional subgroups. The functional microcircuit of the GrCL, thus, needs to be re-examined in the light of the existence of multiple Golgi cell subtypes.

## Physiological significance of the iNC pathway

Optogenetic activation of iNC axons was found to modulate Golgi cell discharge in vitro and in vivo (*Figures 3 and 6*). The most common effect was a short-latency inhibition of spiking (*Figure 3E–H*). In vivo inhibition of spiking could last for tens of milliseconds, most likely due to the iNC neurons' propensity for high-frequency burst firing (*Figure 6*) and to the slow kinetics of the relatively large synaptic conductances at iNC to Golgi cell synapses (*Figure 3*).

Although the iNC axons may represent less than 5% of the afferent fibers in the GrCL (*Hámori et al., 1980*; *Legendre and Courville, 1986*), the ramification of iNC axons, the high divergence of the Golgi cell axon, and the electrical coupling between Golgi cells (*Dugué et al., 2009*) will amplify the potency of the iNC effects on the GrCL network. The anatomical and electrophysiological evidence presented here suggests that the iNC pathway is likely to induce a period of disinhibition in the GrCs. This disinhibition could influence the time-window for MF input integration in GrCs, enhancing GrC excitability and thereby facilitate the activation of PNs (*Chadderton et al., 2004*; *Kanichay and Silver, 2008*; *D'Angelo and De Zeeuw, 2009*). Confirmation of this functional significance will require experimentation in awake animals, as the MF/parallel fiber (PF) pathway is known to be quiescent in anaesthetized animals (*Bengtsson and Jörntell, 2007*; *Wilms and Häusser, 2015*) and no modulation of PN spiking is thus expected by the disinhibition of GrCs (*Figure 6C2*).

No iNC axons were found outside the cerebellar structures in the GlyT2-cre model, and no evidence for iNC contacts on cerebellar GrCs was seen, excluding the possibility of extracerebellar or PF effects on Golgi spiking. However, the glutamatergic projection neurons of the CN collateralising as an 'eNC' pathway that contact the neurogranin-positive Golgi cells (*Tolbert et al., 1976*, *1977*, *1978*; *Hámori et al., 1980*; *Payne, 1983*; *Houck and Person, 2013*, *2015*) may be contacted by local iNC axons. The iNC synapses on the eNC neurons constitute only of a tiny fraction of their synaptic inhibition (*Husson et al., 2014*), but we cannot completely exclude that some facet of the iNC-mediated depression of Golgi cell spiking in vivo may reflect a decrease in excitatory synaptic drive from CN-originating MFs. However, it is unlikely that the short bursts of iNC spikes evoked by our stimulation protocol would result in a long pause in spiking of CN projection neurons as they are extremely resistant to inhibition (*Person and Raman, 2012*; *Chaumont et al., 2013*; *Najac and Raman, 2015*). A short delay in eNC spikes is unlikely to be the principal source of the observed Golgi cell inhibition, unless the Golgi cell activity would be to a large extent determined by CN. Thus, while further work elucidating the functional role of the eNC projection to the cerebellar GrC layer is sorely needed, we conclude here that the inhibition of Golgi cell spiking observed in vivo (*Figure 6*) is mainly caused by a direct inhibition by the iNC axon terminals impinging on Golgi cell dendrites and cell bodies.

A major feature of the iNC projection to the cortex is its divergence: upon relatively localized viral injection in the CN entire lobules may be innervated. Furthermore, single iNC axons traverse long distances along the medio-lateral plane, similarly to the PFs, making numerous contacts on individual Golgi cell dendrites. Such an arrangement could partly explain the synchronization of Golgi cell activity observed along the axis of the lobules (*Vos et al., 1999*). Furthermore, disinhibition of GrCs by iNC axons could enhance and synchronize MF-PF transmission specifically along medio-lateral stripes, possibly contributing to on-beam synchronization of PNs (*Heck and Thach, 2007*). iNC neurons may thus implement refined temporal binding of parasagittal cerebellar modules within a lobule. Intriguingly, while the iNC neurons, like all other CN neurons examined so far, are contacted by PN axons (*Figure 1—figure supplement 1*; compare with *Bagnall et al., 2009*), their intrinsic quiescence (*Uusisaari and Knöpfel, 2010*; *Figure 6*) calls for identification of the sources of synaptic excitation, as they will determine the context within which Golgi cells would be inhibited. Possible candidates include the collaterals of CFs, MFs, and the local axons of CN neurons. While there is no direct evidence either for or against any of these sources, a few earlier works have described inhibition of Golgi cells in response to electrical stimulation of the IO or to sensory stimulation (*Schulman and Bloom, 1981*; *Xu and Edgley, 2008*), suggesting that the IO might be the source of excitatory drive for iNC neurons.

Cerebellar GrC layer gating by Golgi cell network has been postulated for a long time to play a critical role in cerebellar function; however, the absence of experimental tools allowing specific control of the Golgi network during behavior has prevented investigation of this hypothesis. The novel, inhibitory pathway from the CN to the Golgi cells revealed in our present work opens a way for targeted manipulation and analysis of the information gating in the cerebellar granule layer. Furthermore, it is now reasonable to assert that through iNC neurons as well as the collateralization of glutamatergic projection neurons to the GrCL (*Houck and Person, 2015*), the CN hold a key position to control the activity of the cerebellar cortex.

## Materials and methods

### Animals

Experiments were performed on adult mice (p > 30 days; both males and females) of two mouse lines: the GAD-ires-cre (*Taniguchi et al., 2011*) and the GlyT2-cre (*Husson et al., 2014*). These cre-lines, combined with floxed adeno-associated viral (AAVs; see *Table 2* for details) injections into the CN, allowed specific transfection of either GABAergic or glycinergic CN neurons, respectively. In addition, for immunostaining experiments, heterozygous GlyT2-cre mice were bred with GlyT2-eGFP transgenic mice (*Zeilhofer et al., 2005*) and the offspring carrying both GlyT2-cre and GlyT2-GFP genes were transfected as above. Retrograde labeling of the NO neurons (*Figure 1C–D*) was performed in adult wild-type C57BL/6 mice via non-floxed viral injection into the IO. For *Figure 1—figure supplement 1*, adult L7-CHR2-YFP mice (*Chaumont et al., 2013*) were bred with GlyT2-eGFP mouse and double-positive offspring were used for optogenetic experiments. All animal manipulations were made in accordance with guidelines of the Centre national de la recherche scientifique and the Hebrew University's Animal Care and Use Committee.

### Stereotaxic injections

Mice were deeply anesthetized with a mixture of ketamine and xylazine (106 mg/kg and 7.5 mg/kg, respectively) and placed in a stereotaxic frame. Small craniotomies were performed above the CN. The target regions were mostly in the lateral and interpositus nuclei, and the injections were performed unilaterally for immunohistochemical and anatomical protocols, and bilaterally for electrophysiological experiments. A quartz capillary pipette (35- to 40-μm tip diameter) was positioned in the brain at the proper coordinates for CN (1.8–2.2 mm lateral from midline, 3.2–3.4 mm deep, 6.0–6.2 mm from Bregma), and small amount (50–300 nl for electrophysiological experiments, 50–100 nl for immunohistochemical protocols) of viral suspension (summarized in *Table 2*) was slowly pressure-injected either by a hand-held syringe or using a Picospritzer II (General Valve Corporation). In some experiments, additional virus (either non-specific or cre-dependent GFP reporter) was injected in several locations in the cerebellar cortex or into the IO. After the entire volume was injected, pipettes were held at the same position for 10 to 15 min and were then carefully and slowly removed from the tissue, in order to avoid backflow of the viral suspension and unwanted contamination in the cerebellar cortex along the pipette tracts. Animals were closely monitored for 3 days until recovery from surgery and then housed for at least 3 to 4 weeks before being used in experiments, as described below. Throughout this report, we refer to data obtained using the GAD-cre mouse cerebella injected with floxed AAV2/9 virus with mCherry and ChR2 in the CN as

**Table 2**. Summary of the viral constructs used

| Virus | Constructs | Mouse line and injection site | Used in Figures |
|---|---|---|---|
| AAV2/9.EF1.dflox.hChR2 (H134R)-mCherry | Addgene 20,297 | GAD-cre (CN) | 1A, 1E1, 2A, 3A, 3E–I, 4A–H, 6B |
| AAV2/9.EF1a.DIO.eNpHR3.0-EYFP.WPRE.hGH | Addgene 26,966 | GAD-Cre (CN) | 2C, 5D |
| AAV2.1.EF1á.DIO.hChR2(H134R).eYFP | Addgene 20,298 | GlyT2-cre (CN) | 1B, F, 2B, D, 3B–D, 6A, C |
| AAV2.1.CAG-Flex.tdTomato | Allen Institute #864 | GlyT2-Cre x GlyT2-eGFP (CN) | 5A–C, E–F |
| AAV2/9.hSynapsin.EGFP.WPRE.bGH | UPenn AV-9-PV1696 | GAD-cre (CN + Cortex) | 1C, 3A |
| AAV2/9.CAG.Flex.EGFP.WPRE.bGH | Allen Institute #854 | GAD-cre (CCTX) | not shown |

'GAD-cre', and that from GlyT2-cre mouse cerebella injected with floxed AAV2/1 virus with EYFP and ChR2 as 'GlyT2-cre', unless otherwise specified.

## Acute slice preparation

300-µm thick cerebellar slices were cut from the GAD-cre or GlyT2-cre cerebella using the Campden 7000smz oscillating blade microtome and ceramic blades (Campden Instruments, UK). For the experiments performed at HUJI (GAD-cre animals), horizontal slices were prepared at physiological temperature as described previously (*Huang and Uusisaari, 2013*; *Ankri et al., 2014*) and incubated in Solution 1. For the experiments performed at IBENS (GlyT2-cre animals), sagittal slices were prepared using ice-cold Solution 2. After cutting, the slices were rinsed in warm Solution 3 for few seconds before being transferred to a recovery chamber with Solution 4. *Table 3* summarizes the ionic compositions of all experimental solutions. Acute slices of L7-ChR2-YFP × GlyT2-eGFP animals used for *Figure 1—figure supplement 1* were prepared similarly. Notably, in all experiments, the extent of viral transfection was carefully examined in all slices to make sure no unwanted cerebellar structures were labeled.

## In vitro electrophysiological recordings

The slices were incubated for at least 30 min to an hour in physiological temperature (*Huang and Uusisaari, 2013*; *Ankri et al., 2014*) before being transferred to a recording chamber, mounted on an Olympus (BX51WI or BX61WI; Tokyo, Japan) microscope equipped with an epifluorescence illumination pathway (Roper Scientific, Photometrics, Tucson, AZ) and a camera (Vx45; Optronix, Goleta, CA). During experiments, the GAD-cre slices were perfused with room temperature Solution 1 (25–28 °C; flow rate 3 ml/min), and the GlyT2-cre slices were perfused with Solution 4 at physiological temperature (33°C; flow rate 3.5 ml/min). The bicarbonate-buffered solutions (1 and 4) were continuously gassed with 5% $O_2$/95% $CO_2$. Borosilicate glass patch electrodes (resistance 3–12 MΩ)

**Table 3.** Composition of solutions used for slice preparation and experiments

| Experimental details | Solution 1 (in MilliQ water) HUJI Cutting (33°C) and Chamber-perfusion (25–28°C) solutions | Solution 2 (in Volvic water) IBENS Cutting solution Ice-cold | Solution 3 (in Volvic water) IBENS Recovery solution 33°C | Solution 4 (in Volvic water) IBENS Chamber-perfusion solution (33°C) | Intracellular solution IBENS HUJI |
|---|---|---|---|---|---|
| NaCl | 124 | | | 125.7 | 4 |
| K-gluconate | | 130 | | | 140 |
| D-mannitol | | | 225 | | |
| KCl | 3 | 14.6 | 2.3 | 3.3 | |
| Glucose | 20 | 25 | 25 | 25 | |
| $KH_2PO_4$ | 1.2 | | | | |
| $NaH_2PO_4$ | | | 1.25 | 1.25 | |
| $MgSO_4$ | 3.5 | | | | |
| $MgCl_2$ | | | 7.7 | 1.5 | |
| $NaHCO_3$ | 26 | | 25 | 24.8 | |
| $CaCl_2$ | 2 | | 0.51 | 1.6 | 0.5 |
| EGTA | | 2 | | | 5 |
| HEPES | | 20 | | | 10 |
| D-APV (µM) | 20 [in chamber-perfusion solution] | 50 | 50 | 20 | |
| Minocycline (nM) | | 50 | 50 | 50 | |
| Mg-ATP | | | | | 3 |

were filled with intracellular solution (see *Table 3*; pH 7.3, 280 mOsm). For selecting region for patch-clamp experiments, as well as ascertaining that there was no transfection of any cortical neurons, slices were visualized with arc lamp illumination and appropriate filters (for mCherry fluorescence in GAD-cre brains, emission: 605–685 nm, excitation: 530–588; GFP fluorescence: excitation: 473–508 nm; emission: 518–566 nm, for YFP fluorescence in GlyT2-Cre brains, emission: 523–563 nm; excitation: 500–523 nm). Whole-cell patch-clamp recordings, both current clamp and voltage clamp, were acquired using a Multiclamp 700B amplifier (Molecular Devices, Sunnyvale, CA), digitized at 10 kHz (current-clamp experiments) or 50 kHz (voltage-clamp experiments) with USB-6229 acquisition board (National Instruments, Austin, Texas) and low-pass filtered at 2 kHz. Golgi cells were unambiguously identified from other cells in the cerebellar granular cell layer by the size of their soma and their bi-exponential capacitive current (*Dieudonné, 1995*); furthermore, in some experiments using the GAD-cre mice, additional cre-dependent reporter virus was used to label GABAergic Golgi cells in the cortex and was used to guide neuronal selection. Thus, the percentage of s-Golgi cells out of all Golgi cells recorded (*Figure 4*) was biased towards GABAergic Golgi cells.

In the current-clamp experiments, intrinsic electrophysiological properties and synaptic inputs were assessed in Golgi cells either with zero holding current or with negative current injection so that the spontaneously spiking Golgi cells were hyperpolarized to subthreshold voltage values (−55 mV to −60 mV), similar to the resting membrane potential of the not-spontaneous Golgi cells. During voltage-clamp experiments, Golgi cells' holding potential was −50 mV. All experiments were performed in the presence of 20 µM D-2-amino-5-phosphonopentoate (D-APV, Abcam or Sigma Aldrich) and 10 µM 6-cyano-7-nitroquinoxaline-2,3-dione (CNQX, Sigma Aldrich) or 10 µM 2,3-dihydroxy-6-nitro-7-sulfamoyl-benzo[f]quinoxaline-2,3-dione (NBQX, Abcam) to block N-Methyl-D-aspartic acid (NMDA) and α-Amino-3-hydroxy-5-methyl-4-isoxazolepropionic acid (AMPA) receptors, respectively. In some experiments, strychnine (Abcam or Sigma Aldrich) and SR 95531 ('gabazine'; Abcam or Sigma Aldrich) were added to the bath. Regarding *Figure 1—figure supplement 1*, slice perfusion system was similar to what described above. GlyT2-eGFP positive neurons were identified in the CN by epifluorescence and recorded (holding potential -60 mV, intracellular solution as described in *Husson et al., 2014*).

## In vitro optogenetic stimulation

For optogenetic activation of ChR2s in acute GAD-cre slices, whole-field band-pass-filtered Hg-lamp (Oregon Green filter 473–508 nm; ∼5 mW/mm²) was used, while for the GlyT2-cre slices an optical system combining low-numerical aperture (NA) Gaussian beam illumination and fast acousto–optic focusing system with a 473-nm continuous-wave diode-pumped solid-state laser (LRS 0473-00100-03, Laserglow Technologies) was used as a one-photon light source. Small field of view (1.35 µm × 1.08 µm) around the ChR2-expressing fibers was stimulated (stimulation duration 5 ms; inter-stimulation interval 20 s). PN terminals in the CN of L7-ChR2-YFP × GlyT2-eGFP mouse (*Figure 1—figure supplement 1*) were stimulated with 470 nm LED whole-field illumination (Thorlabs, Newton, NJ) with one millisecond duration.

Throughout the work, special care was taken to prevent inadvertent transfection and stimulation of other inhibitory cerebellar interneurons expressing GAD and GlyT2 in our two mouse models. In addition to the precautions taken during the stereotaxic injection procedures, during acute experiments, the slices were carefully and systematically examined before being used for electrophysiological experiments; if unintended labeling was present, the slices were discarded. Finally, before patch-clamping a Golgi cell, the morphology and location of the fibers was carefully examined in order to exclude the possibility of activating parasagittal long-range Lugaro axons.

## In vitro data analysis

Electrophysiological data were analyzed with Igor Pro 6.1 (Wavemetrics, Portland, OR) and MATLAB R2009b (MathWorks, Natick, MA). Statistical analysis was performed using R GNU and MATLAB R2009b or R2012b. Data are presented in the text as mean ± S.D, unless otherwise specified. For statistical significance, Wilcoxon rank-sum, two-tailed Student's t-test (paired or unpaired), F-test, K-S test, and signed-rank tests were used, as applicable, taking into account possible assumptions of normality as mentioned in the results. Spike delay analysis was performed by aligning the last spikes before light stimulation in each trace for each cell and measuring the time to the next spike after light stimulation; the measurements were normalized to the cell's average ISI. For comparing electrophysiological properties of different types of Golgi cells, grand average AP waveforms were

generated for each cell by averaging peak-aligned APs obtained during 1-s voltage sweeps while adjusting the firing frequency to ∼25 Hz with current injection as necessary, and then by averaging these peak-aligned mean waveforms across experiments. $C_m$ was defined as the ratio of membrane resistance ($R_m$) and time constant ($\tau$), estimated from voltage responses (<5 mV) to small hyperpolarizing current steps that did not activate voltage-gated conductances (evidenced by the good single-exponential fits to the voltage responses). AP threshold was defined as the voltage at the time of the main peak in the second derivative of the voltage trace; AP amplitude was measured as the voltage difference between the spike threshold and peak voltage. Spike half-width was measured spike duration at half-amplitude. Spike AHP voltage was measured at the post-spike minimum voltage. AHP time was measured as the time of AHP voltage after threshold, and AHP amplitude was measured as the difference of the AHP voltage and spike threshold voltage. For fair comparison of current-to-firing frequency ratios in different sized cells, the current injection values were normalized to the $C_m$ of each cell. Spike frequency adaptation index was quantified as the relative decrease in instantaneous firing frequency during a 1-s long depolarizing current step (from −65 mV holding level) during which the mean firing frequency was ∼25 Hz. In the pharmacological voltage-clamp experiments (*Figure 3*), the responses were recorded after at least 6 min from the beginning of the perfusion of the drug into the recording chamber to provide the time for the steady-state effect. Time-locking of spikes (*Figure 3—figure supplement 1*) was quantified as the decrease in the normalized $V_m$ variability between subsequent stimulation trials. Decrease in variability, as an index of spike-time locking, was considered statistically significant at the level of 3*SD (see also *Schneider et al., 2014*).

## In vivo electrophysiological recordings and data analysis

Animals were placed in a stereotaxic apparatus (Harvard Apparatus, Halliston, MA). A scalp incision was made along the midline; the skull was cleaned by scraping and by application of hydrogen peroxide. Crus I and II were exposed with a craniotomy, but the dura was not removed to enhance mechanical stability. Commercial tetrodes embedded in a quartz tube (Thomas Recording, Giessen, Germany), gold-plated to reach a 100–200 kΩ impedance, were lowered into the CN. Signals were referenced against a tungsten electrode positioned in saline at the surface of the cerebellar cortex. The light was delivered immediately above the CN via an optical fiber (diameter 200-μm core) connected to CrystaLaser at 473 nm and inserted in a cannula placed above the injection sites in the CN (see 'Stereotaxic injections' paragraph for coordinates). 25- to 50-ms light pulses (45 mW) were delivered at 4 Hz in 3-s bouts separated by 7-s recovery periods; post hoc inspection showed no indication of decreased inhibition during the 3-s bouts, suggesting that these stimulation parameters did not induce cumulative ChR2 inactivation. Signals were acquired using a custom-made headstage and amplifier and a custom-written LabVIEW software (National Instruments, Austin, TX) allowing real-time monitoring of cellular activity. To isolate spikes, continuous wide-band extracellular recordings were filtered off-line with a Butterworth 1 kHz high-pass filter. Spikes were then extracted by thresholding the filtered trace and the main parameters of their waveform extracted (width and amplitude on the 4 channels). The data were hand-clustered by polygon-cutting in 2-dimensional projections of the parameter space using Xclust (Matt Wilson, MIT). The quality of clustering was evaluated by inspecting the auto-correlograms of the units (*Gao et al., 2011*). Golgi cells and Purkinje cells were identified according to most recently published criteria (*Van Dijck et al., 2013*). These criteria provide a simple approach to classify cerebellar units using only a few statistical parameters describing the firing frequency and irregularity of discharge: the mean spike frequency (MSF), the coefficient of variation of the log of the ISI (LCV), and entropy of the ISIs (ENT). We used the boundary values on these parameters defined by *Van Dijck et al. (2013)* to identify Golgi cells (MSF<20 Hz, 0.5<ENT<7.5 and 0.02<LCV<0.25). Golgi cells exhibited an average firing rate (FR) of 8.7 ± 4.2 Hz. To assess the presence of a response to optogenetic stimulations, peri-stimulus time histograms (PSTH) were constructed. Each PSTH was normalized by subtracting the average baseline spike count (before stimulation) and dividing by the baseline standard deviation yielding a Z-score. A modulation of FR in response to optogenetic stimulation was considered significant when the absolute Z-score of a 3-ms bin was higher than 2.5 in at least two time bins in the 50-ms time window following the stimulus. The total time during which the Z-score was significant defined the duration of the inhibition. The latencies (onset and peak) were calculated starting from the beginning of the light-pulse, the peak latency being the point where the Z-score was maximal and the onset latency being defined as the first time point with a significant Z-score.

## Anatomical examination and immunohistochemistry

Animals were deeply anesthetized with intra-peritoneal injection of sodium pentobarbital (50 mg/kg) and perfused through the aorta with ice-cold solution of phosphate buffer saline (PBS; pH 7.4; Sigma, Saint Louis, MO, USA) followed by 50–75 ml of 4% wt/vol paraformaldehyde (PFA; VWR International, Radnor, PA) in PBS. The entire brain was then dissected and post-fixated (3 hr for immunohisto-chemistry, overnight for anatomical examination) in 4% PFA at 4°C before rinsing in PBS. For anatomical confocal imaging, 80-µm sections were cut with a Leica 1000 TS vibratome (Leica Microsystems, Wetzlar, Germany), placed on objective glass, mounted with Immu-Mount (Thermo Scientific, Waltham, MA), and coverslipped with #1.5 glass. For immunohistochemistry experiments, the brains were cryoprotected by equilibration in 30% sucrose wt/vol PBS at 4°C and then cut at −20°C with a Leica CM3050S cryostat (Leica Microsystems, Wetzlar, Germany). Free-floating 80-µm thick parasagittal sections were rinsed in PBS and permeabilized 2 hr at room temperature in 0.4% vol/vol Triton 100-X (Sigma) in PBS. Non-specific sites were saturated by incubation in 0.4% Triton 100-X—1.5% cold fish skin gelatin (Sigma) in PBS at room temperature for 3 hr. Primary antibodies were applied overnight at 4°C in a PBS solution containing 0.1% Triton 100-X—1.5% fish gelatin (mouse GAD65-67 antibody mAB 9A6 (Enzo Life Sciences, Farmingdale, NY) at 1/500 final dilution; chicken GFP antibody (Avès, Oregon, USA) at 1/1000 final dilution; guinea pig VIAAT antibody (Synaptic Systems, Göttingen, Germany) at 1/1500 final dilution; guinea pig GlyT2 antibody (Millipore, Darmstadt, Germany) at 1/1500 final dilution, rabbit Neurogranin antibody (Millipore) at 1/500 final dilution). After rinsing in 0.1% Triton 100-X in PBS, slices were incubated overnight at 4°C with secondary antibodies coupled to 488, 549, or 649 DyLight fluorophores (Jackson ImmunoResearch, West Grove, PA) or Alexa Fluor 555 IgG (Invitrogen, Carlsbad, CA) at 1/500 final dilution in PBS—0.1% Triton 100-X −1.5% cold fish skin gelatin. Slices were finally rinsed with PBS and mounted in Prolong Gold Antifade Reagent (Sigma). For *Figure 1—figure supplement 1*, immunostaining against GFP and VIAAT (same antibodies as above) was performed on paraffin-embedded sections as previously described (*Husson et al., 2014*).

## Image acquisition and analysis

Confocal stacks from immunolabeled cerebellar slices were acquired using an inverted confocal microscope (Leica, SP8) using a 63× oil-immersion objective (NA 1.3). Confocal stacks for anatomical visualizations were acquired with Leica SP5 microscope using 40× (NA 1.25) and 63× (NA 1.5) oil-immersion objectives, with 8- or 12-bit color depth, and with 0.1 µm z-step. The images were acquired for mCherry, EYFP, and GFP fluorescence with excitation lasers and emission filters set to: 561 DPSS laser, 587–655 nm; 488 argon laser, 520–580 nm; GFP, 500–550 nm. Wide-view images (in *Figure 2A1,B1*) were composed by merging tiles of confocal stacks (with 10% overlap and 1 µm z-step). The grayscale background images in 2A1 and B1 were obtained from autofluorescence signals acquired at the same time as the specific fluorescence signals. Morphological features (NC cell body sizes, iNC axonal bouton sizes) were measured using the Fiji image analysis software (*Schindelin et al., 2012*). The soma sizes are given as the major length axis; for axon bouton sizes, as the area of the cross section of each bouton in maximal projection image. To quantify neurogranin and GlyT2-eGFP staining intensities at Golgi cell bodies, z-stacks containing the somata were projected and averaged (z-projection thickness: 6.8 µm). Intensities for each channel were normalized according to the slope of the fit to the logarithmic distribution of their pixel intensity before being retrieved and the ratios were calculated. As the neuropil in cerebellar granule layer is densely labeled in the GlyT2-eGFP mice preventing backtracking individual distal dendrites to their somata in order to attribute them a ratio value, the iNC varicosities in the granular cell layer were included in the statistics only when contacting proximal dendrites and cell bodies. Using GNU R, K-means 2D clustering was performed on mean GlyT2-eGFP vs mean neurogranin data set to cluster the GoC subpopulation.

## Acknowledgements

We thank all members of the Yarom laboratory, and especially Avi Libster and Vitaly Lerner, for helpful discussions and support with technical issues; and Dr Naomi Melamed-Book (ELSC Bio-Imaging unit) for assistance with confocal imaging. We thank Dr Karl Deisseroth and Dr. Edward Boyden for MTAs of adeno-associated-viruses. Research was supported by CNRS, INSERM and ENS, and by Agence Nationale de la Recherche Grants INNET (BL2011) and Edmond and Lily Safra Center for Brain Sciences (ELSC). We are grateful to the IBENS Imaging Facility, which received the support of grants from the

"Région Ile-de-France" (NERF N°2009-44 and NERF N°2011-45), the "Fondation pour la Recherche Médicale" (N° DGE 20111123023), and the "Fédération pour la Recherche sur le Cerveau—Rotary International France" (2011). The IBENS Imaging Facility has also received support implemented by the ANR under the program « Investissements d'Avenir», with the references: ANR-10-LABX-54 MEMO LIFE, ANR-11-IDEX-0001-02 PSL* Research University, and ANR-10-INSB-04-01 France-BioImaging infrastructure. ZH was a recipient of a fellowship from Université Pierre et Marie Curie—ED3C. KP was supported by FRM and Labex MemoLife. LA was supported by Rafik, the Edmond and Lily Safra Center for Brain Sciences (ELSC), and the Benin foundation. MYU was supported by ELSC and CEREBNET (PITN-GA-2009-238686). The work by LA, YY, and MYU was also supported by the Gatsby Charitable Foundation.

## Additional information

### Funding

| Funder | Grant reference | Author |
| --- | --- | --- |
| Gatsby Charitable Foundation | | Lea Ankri, Yosef Yarom, Marylka Yoe Uusisaari |
| Centre National de la Recherche Scientifique | | Zoé Husson, Katarzyna Pietrajtis, Rémi Proville, Stéphane Dieudonné |
| Institut national de la santé et de la recherche médicale | | Zoé Husson, Katarzyna Pietrajtis, Rémi Proville, Stéphane Dieudonné |
| European Commission (EC) | Marie Sklodowska-Curie Actions (MSCA) / CEREBNET: PITN-GA-2009-238686 | Marylka Yoe Uusisaari |
| Israel Science Foundation (ISF) | | Yosef Yarom |
| Agence Nationale de la Recherche | BL2011 | Zoé Husson, Katarzyna Pietrajtis, Rémi Proville, Stéphane Dieudonné |
| Selim and Rachel Benin Scholarship Fund | | Lea Ankri |
| Ecole Normale Superieure | | Zoé Husson, Katarzyna Pietrajtis, Rémi Proville, Stéphane Dieudonné |
| Hebrew University of Jerusalem | Edmond and Lily Safra Center for Brain Sciences | Lea Ankri, Marylka Yoe Uusisaari |
| Région Ile-de-France | NERF N°2009-44 | Stéphane Dieudonné |
| Région Ile-de-France | NERF N°2011-45 | Stéphane Dieudonné |
| Université Pierre et Marie Curie (UPMC) | Fellowship | Zoé Husson |
| Ford Foundation | Roberto Marinho Foundation - FRM | Katarzyna Pietrajtis |
| Labex | MemoLife | Katarzyna Pietrajtis |

The funders had no role in study design, data collection and interpretation, or the decision to submit the work for publication.

### Author contributions

LA, ZH, KP, RP, MYU, Conception and design, Acquisition of data, Analysis and interpretation of data, Drafting or revising the article; CL, Analysis and interpretation of data, Drafting or revising the article; YY, Conception and design, Drafting or revising the article; SD, Conception and design, Analysis and interpretation of data, Drafting or revising the article

### Author ORCIDs

Rémi Proville, http://orcid.org/0000-0002-5829-1344

## Ethics

Animal experimentation: All animal manipulations were made in strict accordance with guidelines of the Centre National de la Recherche Scientifique and the Hebrew University's Animal Care and Use Committee (Ethic permission # NS-12-12505-4).

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
