## [Decision Letter]

Thank you for sending your work entitled “A novel inhibitory nucleo-cortical circuit controls cerebellar Golgi cell activity” for consideration at *eLife*. Your article has been favorably evaluated by Eve Marder (Senior editor), a Reviewing editor, and three reviewers.

The Reviewing editor and the reviewers discussed their comments before we reached this decision, and the Reviewing editor has assembled the following comments to help you prepare a revised submission.

This study uses an extensive range of histochemical, electrophysiological and optogenetic techniques, both in vitro and in vivo, to describe the anatomical and functional properties of an inhibitory projection from the cerebellar nuclei to the cerebellar cortex. The main finding is that a specific subpopulation of GlyT2+/GAD+ neurons located in the cerebellar nuclei, innervate a distinct subpopulation of Golgi cells. The axons of the CN neurons co-release GABA and Glycine and can either pause or drive the firing of the Golgi cells. Both mixed GABAergic/Glycinergic and pure GABAergic Golgi cells are targeted by the feedback projection, while pure glycinergic Golgi cells are not.

The referees agreed that the study is of high quality, and that the results are important because they establish the presence of nucleo-cortico feedback, which is likely to have significant functional implications for certain regions of the cerebellar cortex.

However, there were some concerns about the experimental procedures used, which will require additional experiments and/or clarification. These include more details about the specificity of transfection protocols; the potentially weak effect on circuit function; missing details about the inputs to iNCs and their outputs to Golgi cells, paying attention to potential local circuit explanations for the observed effects on GC excitability; and missing details about the inputs to iNCs and their outputs to Golgi cells/ and potentially other inhibitory and excitatory neurons. Furthermore, throughout the manuscript the impact of the findings is overstated, and the level of scholarship should be improved, particularly with regard to citations. Below is an outline of the concerns that need to be addressed.

1) Is this really a disinhibition circuit? Although the authors provide a nice description of the synaptic contacts onto Golgi cells, it remains unclear whether excitatory cells are also contacted, either in the cerebellar cortex or elsewhere. The authors may or may not have some data already that could address this. In any case, this issue should be discussed.

2) A major assumption in the manuscript is that the identified iNCs receive Purkinje cell inhibitory input. This is critical to demonstrate, as an alternative interpretation to that presented in this manuscript is that iNC cells are essentially ectopic Golgi cells, receiving mossy fiber collateral input but no descending PC input.

3) This appears to be a sparse and weak pathway and it is not clear how significant the functional impact could be, for two reasons:

A) Although optogenetic stimulation is very powerful for stimulating all synapses of a specific input simultaneously, the responses still appear rather weak (Figure 3). Therefore, it would be useful to know what the holding potential was in the voltage clamp experiments, and what the driving force for Cl- was. That would allow an estimate of the total evoked synaptic inhibitory conductance. These parameters do not seem to be provided in the manuscript. In addition, the measured inhibitory responses in Golgi cells are an overestimate of their functional impact because it is unlikely that all these inhibitory inputs are active simultaneously during behavior.

B) The in-vivo electrophysiology experiments show a strong inhibitory effect on Golgi cell firing. It is not clear however, whether this is due to a direct inhibition of Golgi cells, or an indirect effect by inhibition of excitatory input that contributes to driving the Golgi cells (see also point 1 above).

Additional experiments to address these comments are not essential but would be potentially welcome. In any case the authors should discuss the functional importance of this pathway in a balanced way, and in particular the caveats of the in vivo experiments.

4) How can the authors be sure that the inhibitory currents evoked by ChR2 arise solely from iNC and not from other inhibitory interneurons? It is possible that the injection pipette released some virus on its way down to the cerebellar nuclei and infected other cell types (e.g. Golgi cells or Lugaro cells), which could be the source of the inhibition. The specificity and extent of the ChR2 expression needs to be verified, to rule out this possibility. In particular, were Purkinje neurons ever labeled in the GAD-cre mice? If not, why not? If so, how was this dealt with? And given that pipettes used for viral injection must pass through the granule cell layer when targeting CN, were Golgi cells ever transfected in either GAD-cre or GlyT2-cre lines? It should be possible to tell whether Golgi cells or Lugaro cells are infected: their axonal arbors are morphologically distinct, and they are also distinct from the iNC projections. It should also be possible to record far away from the injection site. It is likely that the authors have done this but they should clarify their injection/recording strategy.

5) The section relating to Figure 5 is problematic in places. The term “associated” used to link Neurogranin expression with GABAergic neurons is somewhat misleading. In the Simat paper, some mixed neurons do not express Neurogranin (type 2 and 3, respectively 3-5% and 10%), although they release GABA (GAD67+). Thus, a significant proportion of the neurons classified as pure glycinergic could be mixed. While it is possible that these neurons match 3% of “pure glycinergic” neurons contacted by iCN projections, based on the Simat estimates, the proportion should be higher than 3%. Moreover, the ratio of Neurogranin/eGFP-GlyT2 is rather an indirect way of assessing the identity of a cell given the level of fluorescence is influenced by many experimental parameters. Indeed, the cells expressing Neurogranin at 2x higher levels than the level of GlyT2-eGFP, which are classified as GlyT2 negative, could still secrete a substantial amount of Glycine. This makes the conclusion that “these results demonstrate that iNC fibres contact almost exclusively the purely GABAergic Golgi cells” uncertain. The wording should be altered to reflect these limitations.

6) The method used to measure synchrony is unusual and indirect. Please use a more conventional measure of synchrony.

7) Understanding the impact of this new feedback connection is limited without a basic characterization of its short-term plasticity. Does it depress or does it facilitate? The short-term plasticity would add significant functional insight. If the authors already have this data they should provide it.

8) As it is a key finding in the study, it is critical to provide quantification for the double labeling experiments in the last paragraph of the subsection headed “Nucleo‐cortical projection neurons have a mixed GABA‐glycine phenotype” and Figure 2.

9) The differences in spontaneously firing and silent Golgi cells described in the subsection headed “iNC fibers inhibit a subpopulation of Golgi cells with different electrophysiological properties” seem out of place in this manuscript given that the iNC responsive Golgi cells amount to only 35% of the spontaneously firing population. Either this should be removed, or better justified.

10) The Abstract and beginning of the Introduction should be rewritten. The authors refer to an “orthodox view” that information flows predominantly from cortex to nucleus within the cerebellum. This statement is then immediately contradicted by numerous (although incomplete, see below) citations, many 30-40 years old (e.g., Tolbert, Hamori, Batini).

11) Given that the in vivo tetrode recording experiments of Figure 6 showed no modulation of Purkinje cell activity, many of the prominent claims of the manuscript (e.g. “… the iNC pathway places the CN in a position to control the signal transmission between the MFs and PNs and thereby to modulate activity in the cerebellar cortex in response to sensory‐motor information” and “Our data strongly suggest that the CN contributes to the functional recruitment of the cerebellar cortex by disinhibiting the mossy fiber-parallel fiber pathway via the iNC projection”, etc.) seem overstated.

Minor comments:

A) In Figure 1 and in the first paragraph of the Results, “arrow” and “arrowhead” seem to be switched. B2 is also not mentioned in the figure legend.

B) Distributions were compared using an F-test. However, this test is sensitive to non-normality (Figure 1, the red distribution is possibly not normal). A non-parametric test such as the Kolmogorov Smirnov test would be more appropriate in Figures 1 and 2.

C) Figure 21 indicates a significant difference between the two distributions, but the text (legend and main text) states that “The swelling labelled in the two mouse lines were identical in size”. Could the very small difference between the two distributions (around 200nm) be simply due to the different wavelengths used to image mCherry or YFP signals? Bin sizes (and bin centres) also look different between the two distributions.

D) In the Results, it is stated that the immunostaining revealed unequivocally that inhibitory NC projections have a dual neurotransmitter phenotype. However, given there is only one example of each, and no quantification, this statement seems too strong.

E) The firing phenotype observed here (silent GoC versus firing GoC) could be discussed in relation to the Hull et al. paper, 2013. In this paper, hyperpolarization of GoC resulted in a long term increase in the firing rate. Could the hyperpolarizing connection from iCN have induced a long lasting, more depolarized state observed here as the firing phenotype (while cells receiving no inhibitory inputs do not present this phenotype)?

F) In the subsection headed “iNC fibers inhibit a subpopulation of Golgi cells with different electrophysiological properties”, the authors state: “ns-Golgi cells showed significantly larger variability in the values than the s-Golgi cells”. Which values?

---

## [Author Response]

*1) Is this really a disinhibition circuit? Although the authors provide a nice description of the synaptic contacts onto Golgi cells, it remains unclear whether excitatory cells are also contacted, either in the cerebellar cortex or elsewhere. The authors may or may not have some data already that could address this. In any case, this issue should be discussed*.

We agree that the significance of the main finding of our work—that the iNC axons target the GABAergic Golgi cells in a profuse and strong manner—may have led to the lack of discussion regarding other possible targets of the iNC axons. We have now included in the manuscript our considerations regarding this issue as follows:

First, no labeled axons were ever seen projecting outside the cerebellum in GlyT2-cre mouse injected in the cerebellar nuclei, ruling out extracerebellar targets of the iNC axons. Second, the granule cells (GrCs) do not seem to be targeted by the iNC projections, as we do not find evidence of iNC fibers entering the glomeruli. Furthermore, the density of fibers and varicosities is much lower than would be expected for a pathway targeting granule cells (compared with Golgi axon density in GrCL), and thus, if iNC-GrC contacts existed, they would be found on a minority of the GrCs. These possibilities are now mentioned in the Discussion (in the subsection headed “Bursting activity in iNC neurons and their inhibitory effect on Golgi cell activity in vivo”).

It should be mentioned that while the vast majority of iNC axonal varicosities were found in apposition to neurogranin-positive Golgi cells, very rarely some of them appeared to face unlabeled cellular targets. Unfortunately, we cannot make strong claims regarding the identity of the possible postsynaptic targets of these “orphan” terminals, as we cannot exclude the existence of Golgi cells negative for both GlyT2-eGFP and neurogranin (or expressing them at levels that were too weak to be detected; see also our answer #5).

Finally, we have previously shown that the local axons of iNC cells contact the glutamatergic projection neurons of the CN (48); we will discuss the possible significance of this pathway to the Golgi cell inhibition with our response to 3B.

*2) A major assumption in the manuscript is that the identified iNCs receive Purkinje cell inhibitory input. This is critical to demonstrate, as an alternative interpretation to that presented in this manuscript is that iNC cells are essentially ectopic Golgi cells, receiving mossy fiber collateral input but no descending PC input*.

First, we do not see PN inhibition onto iNC cells as a major assumption in the paper, and we were not dealing with this question at all in the text. PNs have been proposed to innervate all CN cell types (81; 97) including glycine-containing neurons (25). We now provide direct optogenetic evidence for a functional inhibitory connection from PN onto iNC neurons in a supplementary figure (please see Figure 1—figure supplement 1 and Results).

As for the possibility that iNC neurons were “ectopic Golgi cells”, the described iNC cells constitute about one third of the CN neurons (48). Furthermore, iNC neurons do not appear to share any physiological or morphological characteristics with Golgi cells (Figure 6, Figure 6; [89]), except maybe their dual neurotransmitter content. Demonstration of a contact by PN axons confirms the distinction, as Golgi cells do not receive Purkinje cell inputs in the cortex.

*3) This appears to be a sparse and weak pathway and it is not clear how significant the functional impact could be, for two reasons*:

*A) Although optogenetic stimulation is very powerful for stimulating all synapses of a specific input simultaneously, the responses still appear rather weak (*Figure 3*). Therefore, it would be useful to know what the holding potential was in the voltage clamp experiments, and what the driving force for Cl- was. That would allow an estimate of the total evoked synaptic inhibitory conductance. These parameters do not seem to be provided in the manuscript. In addition, the measured inhibitory responses in Golgi cells are an overestimate of their functional impact because it is unlikely that all these inhibitory inputs are active simultaneously during behavior*.

First, we apologize for the lack of description of the experimental conditions for the recordings of iNC IPSCs in Golgi cells. Recordings were performed in physiological chloride conditions and at a holding potential of -50 mV, as now mentioned in the Materials and methods section.

Second, even though the IPSPs seen in response to stimulation are not very large in amplitude, they seem to be strong enough. The most likely reason is large conductance activated by iNCs, resulting in shunting inhibition (rather than inhibition by hyperpolarization). While we have not experimented on the synaptic properties in detail, employing GHK-based calculations of peak slope conductance, including Cl/HCO_3_ permeability ratios give 1.9 ± 1.3 nS (corresponding to 8.5 ± 5.9 nS in symmetrical chloride conditions; Houston CM et al., J.Neurosci. 2009). This is 6 times the value for Golgi-Golgi connections (0.33 nS; Hull & Regehr, 2012) and amounts to the simultaneous activation of about 300 GABA_A_R-associated channels. The effectiveness of the iNC inhibition on blocking Golgi cell spiking is enhanced by the long time course of the inhibitory conductance. These calculations are now added to the manuscript (in the subsection headed “Nucleo-cortical fibers inhibit Golgi cell activity“).

Third, the reviewers suggest that we might stimulate a large number of iNC axons synchronously, hence overestimating the weight of unitary connections. This is not likely for the following two reasons. First, in the voltage-clamp experiments, stimulation was targeted at a single identified fiber by illuminating a very small area (order of 2 µm^2^), making it unlikely that great many individual axons were activated.

Furthermore, as is evident in the images shown in Figure 5, each Golgi cell seems to be innervated only by one or at most two individual axons in our material, each of the axons making numerous boutons while climbing along the Golgi dendrites. Thus, even wide-field illumination should not activate more than a few axons, excluding gross exaggeration of the iNC effect in slices.

B) The in-vivo electrophysiology experiments show a strong inhibitory effect on Golgi cell firing. It is not clear however, whether this is due to a direct inhibition of Golgi cells, or an indirect effect by inhibition of excitatory input that contributes to driving the Golgi cells (see also point 1 above).

*Additional experiments to address these comments are not essential but would be potentially welcome. In any case the authors should discuss the functional importance of this pathway in a balanced way, and in particular the caveats of the in vivo experiments*.

These issues were also discussed in relation to question 1. First, the inhibitory effect observed during in-vivo experiments was not qualitatively different from what was seen in acute slices, suggesting that additional, long-range circuit mechanisms are not necessary to presume in order to explain the results.

Second, as was mentioned earlier, the iNC cells do not project outside the cerebellum, and strong connections between iNC axons and granule cells are highly unlikely. Thus, iNC activation should not have any effect on Golgi cell spiking, neither via extracerebellar mossy fibers nor via the parallel fibers.

Third, iNC neurons have been shown to project onto CN glutamatergic principal neurons (48), and indeed, during the time we were working on the revision, Houck and Person published anatomical work describing that at least some of the glutamatergic projection neurons of the CN collateralize into the cerebellar cortex, where they form rosette-like terminals that contact both Golgi and granule cells. Thus it could be conceivable that inhibition of spiking in these excitatory CN neurons would result in decrease in Golgi cell spiking. However, the glutamatergic projection neurons are extremely strong pacemakers and resistant to even the massive, tonic PN inhibition (Person & Raman, Nature 2012; Figure 4). The iNC inhibition of these cells is however expected to be significantly weaker than the PN-originating inhibition. Thus, while in principle possible, it seems unlikely that the short burst of iNC IPSPs on glutamatergic CN neurons would reduce the excitatory drive to Golgi cell to result in the observed pauses in Golgi cell spiking in vivo. Notably, this possibility would mean that the Golgi cell activity would have to be to a large extent determined by CN activity.

Furthermore, in 8/18 (44%) of the Golgi cells inhibited in the in vivo experiments, the pause in spiking was followed by a rebound increase in spiking probability in the 100ms following the end of the light pulse (starting 47.6 ± 15.3 ms after the end of the light pulse lasting 9.8 ± 6.6 ms; considered significant at z=2.5 level). Such an increase is likely to result from delayed Golgi cell firing following the direct afferent inhibition, and is not expected to occur if the Golgi cell pause was produced by a decrease in incoming excitation. Overall, while it is impossible to exclude presently uncharacterized circuit mechanisms influencing Golgi cell spiking in the in vivo experiments, our observations are most easily explained (and experimentally supported) by a direct iNC input rather than an indirect effect on Golgi cells excitatory drive. As mentioned earlier, we have now added discussion of these possibilities in the subsection headed “Physiological significance of the iNC pathway” of the manuscript.

*4) How can the authors be sure that the inhibitory currents evoked by ChR2 arise solely from iNC and not from other inhibitory interneurons? It is possible that the injection pipette released some virus on its way down to the cerebellar nuclei and infected other cell types (e.g. Golgi cells or Lugaro cells), which could be the source of the inhibition. The specificity and extent of the ChR2 expression needs to be verified, to rule out this possibility. In particular, were Purkinje neurons ever labeled in the GAD-cre mice? If not, why not? If so, how was this dealt with? And given that pipettes used for viral injection must pass through the granule cell layer when targeting CN, were Golgi cells ever transfected in either GAD-cre or GlyT2-cre lines? It should be possible to tell whether Golgi cells or Lugaro cells are infected: their axonal arbors are morphologically distinct, and they are also distinct from the iNC projections. It should also be possible to record far away from the injection site. It is likely that the authors have done this but they should clarify their injection/recording strategy*.

This is indeed a critical issue, and we have spent a lot of effort since the early stages of the work to ascertain that unspecific stimulations did not occur. Most importantly, throughout the work, we have taken special care to avoid inadvertent transfection of other inhibitory cerebellar neurons. We injected only very small amounts of viral suspension to restrict the transfected volume and the injection pipettes were removed slowly 10-15 minutes after the injection was finished to avoid backflow along the pipette tract. Furthermore, we aimed for a ventral locus of injection in the CN to increase the distance to the cortex. During acute experiments, the slices were carefully and systematically examined before being used for electrophysiological experiments; if unintended labeling was present, the slices were discarded. Also, the recordings were never performed in the parts of the cerebellar cortex adjacent to the CN.

In vivo recordings were performed in the hemispheric dorsal Crus lobules, while injections were being performed in the interposed nucleus through the vermis/paravermis. The optic fiber was placed just above the injection site, in the cerebellar nuclei. This configuration would prevent any unspecific optogenetic stimulation.

Importantly, as was noted by the reviewers, the morphology of Lugaro and Golgi axons is very different from what we have now described for the iNC axons. While not within the scope of the present study, to illustrate the difference, we measured the axonal bouton areas of Golgi and Lugaro cells (from material outside this study where the virus was directly injected into the cortex; n = 100 boutons for both Lugaro and Golgi axons). As evident in Figure 7, the iNC terminals can clearly be differentiated from both Lugaro and Golgi terminals. Thus, before the patch-clamp experiments, the morphology and location of labeled fibers was carefully examined in order to exclude the possibility of activating parasagittal long-range Lugaro axons or nearby Golgi axons.

Author response image 1.Comparison of the axonal swelling size between Lugaro, Golgi and iNC axons reveals that the iNC boutons are significantly larger, allowing reliable differentiation between the different axons even in acute slices. Lugaro and Golgi confocal images obtained from cerebellar slices from GlyT2-cre and GAD-cre mice, where floxed EYFP or GFP-expression virus was injected into the cerebellar cortex. iNC image from material used in the present study.**DOI:**
http://dx.doi.org/10.7554/eLife.06262.016

Unintended PN transfection could possibly happen retrogradely via their axons projecting into the CN. However, in line with the fact that retrograde transfection is much less effective than transfection through somatic membranes, we nearly never saw such retrogradely labeled PNs with our limited-range injections (and only after very long transfection times). In any case, such inadvertent transfections do not pose a problem as PNs do not contact Golgi cells. As for the significance of the possibility of accidentally activating PN axons that would inhibit the glutamatergic CN projection neurons possibly sending collaterals to the cortex and driving Golgi cells, we ask the reviewers to refer to our answers to questions 1 and 3B. Finally, transfection of PN was never observed in the GlyT2-cre background, which was used for in-vivo experiments.

By combining all these precautions, we are confident about the reliability of the data we present, as it is highly consistent and uniform among the experiments and between the two laboratories, despite some differences in our experimental procedures.

The meticulous examination of putative contaminations and precautions we have undertaken are now stressed more in details in the revised manuscript (Methods section).

*5) The section relating to*
Figure 5
*is problematic in places. The term “associated” used to link Neurogranin expression with GABAergic neurons is somewhat misleading. In the Simat paper, some mixed neurons do not express Neurogranin (type 2 and 3, respectively 3-5% and 10%), although they release GABA (GAD67+). Thus, a significant proportion of the neurons classified as pure glycinergic could be mixed. While it is possible that these neurons match 3% of “pure glycinergic” neurons contacted by iCN projections, based on the Simat estimates, the proportion should be higher than 3%. Moreover, the ratio of Neurogranin/eGFP-GlyT2 is rather an indirect way of assessing the identity of a cell given the level of fluorescence is influenced by many experimental parameters. Indeed, the cells expressing Neurogranin at 2x higher levels than the level of GlyT2-eGFP, which are classified as GlyT2 negative, could still secrete a substantial amount of Glycine. This makes the conclusion that “these results demonstrate that iNC fibres contact almost exclusively the purely GABAergic Golgi cells” uncertain. The wording should be altered to reflect these limitations*.

First of all, we do not claim that 3% of the iNCs form synapses on “pure glycinergic” neurons, but that 3% of the iNC were seen to contact neurons which expressed some level of GFP in the GlyT2-GFP mice, and therefore probably are not “pure GABAergic” Golgi cells. Regarding Golgi cell subtypes as described by [75], it should be noted that all neurogranin-positive Golgi cells express GAD67 (while the opposite is not true). Thus, the statement that neurogranin is “associated” with GAD expression is in our opinion fully justified. The neurogranin-positive cells that carry 97% of detected iNC contacts can therefore belong to either the purely GABAergic neurons (Simat’s type 4, 15% of all cells) or the mixed GABA-glycinergic Golgi cells (Type 1, 65%). The remaining 3% of iNC axons could contact either Simat’s types 2, 3 or 5. Notably, we cannot distinguish between these “Simat” classes in our method.

In any case, we thank the reviewers for their comments, which prompted us to clarify the analysis of our data and to improve their presentation. Thus, we now classify the GoCs based on the GlyT2-GFP only, and not the ratio. Unlike the classification of Golgi cells done by Simat, we are not differentiating Golgi cells to strict classes. Instead we are showing preferential targeting of cells residing on the extreme end of a continuum. The Golgi cell population can be divided into at least two groups as attested by the bimodal distribution of GFP expression. The larger cluster (labeled green in Figure 5) of cells displays strong GFP expression while the second cluster (blue) displays low to background levels of GFP expression. All the cells which are strongly contacted by iNC axons (i.e., have more than 1-4 contacts) fall in the second cluster of low-GFP neurons. Importantly, the most strongly contacted cells in this group (labeled with darker reds in Figure 5) have relatively high neurogranin expression. This group most likely corresponds to Simat’s type 4 and can reasonably be called “GABAergic Golgi cells”, although some of them do seem to be stained for GFP. As expected, many cells of the blue cluster were not found to receive iNC contacts in our experiments, since we transfected only a small fraction of all iNC neurons in the CN.

We also found a few iNC contacts on a small number of Golgi cells belonging to the first cluster, displaying high GlyT2-eGFP expression (Figure 5, yellow markers at the top left corner, 8 out of 40 iNC-contacted Golgi cells). However, iNC fibers never formed an extensive articulation with these cells, that also always displayed only weak neurogranin expression.

To clarify the conclusions we have reached based on our data, we have added a schematic drawing (Figure 5) to the paper.

*6) The method used to measure synchrony is unusual and indirect. Please use a more conventional measure of synchrony*.

The word “synchrony” is rather confusing in this context, since the results are not showing synchronicity of several cells, but a time-lock of the spikes in one cell. The index analysis we utilized is not that novel as it has been used in previous literature (see Figure 1 in [73]). However, these results were found post-hoc in 7 Golgi cells, after we had completed the data acquisition, and thus the experiments were not designed to address this issue. Therefore, we unfortunately have too few repetitions to conduct a detailed phase-response analysis. As it is not critical for the paper, we decided to remove this panel from the figure and only mention it as a supplementary figure (please see the subsection headed “Nucleo-cortical fibers inhibit Golgi cell activity” and Figure 3—figure supplement 1).

*7) Understanding the impact of this new feedback connection is limited without a basic characterization of its short-term plasticity. Does it depress or does it facilitate? The short-term plasticity would add significant functional insight. If the authors already have this data they should provide it*.

While we do agree that this information would be of interest, we have unfortunately not performed the experiment.

*8) As it is a key finding in the study, it is critical to provide quantification for the double labeling experiments in the last paragraph of the subsection headed “Nucleo‐cortical projection neurons have a mixed GABA‐glycine phenotype” and*
Figure 2.

Quantification of the double labeling experiments shown in Figure 2 were performed and results are now included in the text.

*9) The differences in spontaneously firing and silent Golgi cells described in the subsection headed “iNC fibers inhibit a subpopulation of Golgi cells with different electrophysiological properties” seem out of place in this manuscript given that the iNC responsive Golgi cells amount to only 35% of the spontaneously firing population. Either this should be removed, or better justified*.

Most importantly for the justification of the division into two Golgi groups, the iNC-originating inhibition was never seen in the ns-Golgi cells that clearly differ in their electrophysiological signature from the s-Golgi cells. Combining this observation of connectivity difference with electrophysiological differences is strengthening the notion that the Golgi cells are composed of several distinct cell groups. While not the main focus of this study, because of the common view in current literature is of Golgi cells as a homogenous group of cells, we believe that it is highly relevant to stress the differences found in the present work.

The relatively small percentage of patched s-Golgi cells where we could demonstrate an inhibitory effect of the iNCs is mostly due to experimental limitations, as a small fraction of all iNC cells in a given animal are transfected in our conditions. Indeed, the injections were as restricted as possible to prevent transfection of cortical interneurons (see above). Such partial transfection will result in a large number of false negative s-Golgi cells. We believe that in ideal conditions, in which all iNC cells would be transfected, 100% of the s-Golgi cells would be contacted.

*10) The Abstract and beginning of the Introduction should be rewritten. The authors refer to an “orthodox view” that information flows predominantly from cortex to nucleus within the cerebellum. This statement is then immediately contradicted by numerous (although incomplete, see below) citations, many 30-40 years old (e.g., Tolbert, Hamori, Batini)*.

The Abstract and Introduction were rewritten in a more logical and appropriate manner. We apologize for having submitted an inconsistent text.

*11) Given that the in vivo tetrode recording experiments of*
Figure 6
*showed no modulation of Purkinje cell activity, many of the prominent claims of the manuscript (e.g. “… the iNC pathway places the CN in a position to control the signal transmission between the MFs and PNs and thereby to modulate activity in the cerebellar cortex in response to sensory‐motor information” and “Our data strongly suggest that the CN contributes to the functional recruitment of the cerebellar cortex by disinhibiting the mossy fiber–parallel fiber pathway via the iNC projection”, etc.) seem overstated*.

As the in vivo experiments were performed under anesthesia, we expected no modulation of the Purkinje cells firing, as granule cells are not active under such conditions (42; 94). However, we agree with the reviewers that we perhaps overstated the impact of iNC activity on mossy fiber integration in the granule cell layer, as we do not provide direct evidence on this issue. We have revised the relevant parts of the Discussion.

Minor comments:

*A) In*
Figure 1
*and in the first paragraph of the Results, “arrow” and “arrowhead” seem to be switched. B2 is also not mentioned in the figure legend*.

Appropriate modifications were made in the text and in the figure.

*B) Distributions were compared using an F-test. However, this test is sensitive to non-normality (*Figure 1*, the red distribution is possibly not normal). A non-parametric test such as the Kolmogorov Smirnov test would be more appropriate in*
Figures 1 and 2.

We thank the reviewers for the correction. We conducted the Kolmogorov-Smirnov test and the current results are: GAD vs. NO: p = 0.0082; GAD vs. Gly: p < 0.0001; NO vs Gly: p < 0.0001 (in the subsection headed “Nucleo-cortical projection neurons have a mixed GABA-glycine phenotype”).

*C)*
Figure 2
*indicates a significant difference between the two distributions, but the text (legend and main text) states that “The swelling labelled in the two mouse lines were identical in size”. Could the very small difference between the two distributions (around 200nm) be simply due to the different wavelengths used to image mCherry or YFP signals? Bin sizes (and bin centres) also look different between the two distributions*.

Due to the large number of measurements used for the statistical analysis, we observed a statistical difference even though the difference between the two distributions is very small. As noticed by the reviewer, this small difference could be indeed explained by the two different fluorophores used for imaging with different wavelengths.

After reconsidering the issues, we have decided to replace the histograms in Figure 2 by cumulative probability plots, which in our view give more accurate and clear view on the distributions. The midpoints of GAD and GlyT2 populations are at 1.95 and 1.77 µm^2^, and for GrCL and ML populations, 1.77 and 2.0 µm^2^, corresponding to approximately 250 nm^2^ or 50 nm diameter difference in measurements. The attached Q-Q plots (Figure 8) comparing the distributions between GAD and GlyT2, and GrCL and ML boutons, show very little deviation from unity line (slopes 0.86 and 0.97) and should be sufficient as a demonstration that this tiny difference is unlikely of having profound significance.

Author response image 2.Comparison of the quantiles of cu- mulative distributions of iNC axon swellings obtained with GAD-cre vs. GlyT-cre models (top) and from granule cell layer vs. molecular layer (bottom; GAD-cre and GlyT2-cre models pooled). The near-perfect match with unity lines demonstrates near-identity.**DOI:**
http://dx.doi.org/10.7554/eLife.06262.017

*D) In the Results, it is stated that the immunostaining revealed unequivocally that inhibitory NC projections have a dual neurotransmitter phenotype. However, given there is only one example of each, and no quantification, this statement seems too strong*.

Percentages of co-labelled varicosities in both GAD-cre and GlyT2-cre were calculated and added in the text (in the subsection headed “Nucleo-cortical projection neurons have a mixed GABA-glycine phenotype”). For both GAD and GlyT2 models, the amount of co-labeling was very high (94.6 ± 6.2 % and 93.9 ± 5.0 %, respectively).

E) The firing phenotype observed here (silent GoC versus firing GoC) could be discussed in relation to the Hull et al. paper, 2013. In this paper, hyperpolarization of GoC resulted in a long term increase in the firing rate. Could the hyperpolarizing connection from iCN have induced a long lasting, more depolarized state observed here as the firing phenotype (while cells receiving no inhibitory inputs do not present this phenotype)?

The idea from [46] revolves around tonic hyperpolarization. A long-lasting (at least 20 s) inhibitory activity is needed in order to affect baseline firing frequency. However, the iNC -mediated inhibition we describe here is most definitively of a phasic kind, due to the bursting nature of the iNC cells, and thus unlikely to be related.

The simplest explanation for the difference in spontaneous spiking between s-Golgi cells and ns-Golgi cells is that they are different in their intrinsic electrical properties, a fact also corroborated by our findings (Figure 4). Notably, in the Hull et al. work (2013), it was explicitly stated that the shift in excitability is not associated with AP waveform changes whereas in our work, one of the clear difference between the s-Golgi cells and ns-Golgi cells is the AP half-width.

*F) In the subsection headed “iNC fibers inhibit a subpopulation of Golgi cells with different electrophysiological properties”, the authors state: “ns-Golgi cells showed significantly larger variability in the values than the s-Golgi cells”*. *Which values?*

This section has been significantly revised.